# Dual-Stream Neural Fractional Operator for Nonstationary Multivariate Time Series Forecasting

## Abstract

Long-term, multivariate time-series forecasting is vital for domains such as energy systems, finance, and weather prediction, where accurately modeling intricate patterns can yield significant performance gains. However, many existing models struggle with the inherent non-stationarity of real-world data, where distribution shifts can vary both within and across series, leading to suboptimal long-horizon forecasts. While techniques like normalization and decomposition have been applied to learn more nuanced features, they often rely on restrictive assumptions. To overcome these limitations, we propose DualFrac, a dual-stream system is built on stacked neural fractional operators, each performing fractional-domain, time-varying transformations with interwoven decomposition to extract non-stationary sub-components for weaving the target signals. By producing a hierarchy of sub-forecasts that are progressively aggregated, our model effectively captures both intra-series and inter-series dependencies in a non-stationarity-aware manner. Extensive experiments show that our approach achieves state-of-the-art (SOTA) performance, surpassing recent decomposition-based and transformed domain models, further validating its robustness and effectiveness.

## 1 Introduction

Long-term time series forecasting (LTSF) underpins critical applications in finance, transportation, and climate science. A key challenge arises from *non-stationarity*, where inter-variate coupling (Tajeuna et al., 2022), irregular events, and chaotic intermittency create evolving dynamics that standard deep models fail to capture. While recent advances, ranging from Transformer architectures (Wu et al., 2022; Zhou et al., 2021b) to frequency-domain approaches (Wang et al., 2025; Yi et al., 2023)—have improved accuracy, they often rely on normalization or oversimplified assumptions about underlying dynamics (Liu et al., 2023a;c), which may degrade non-stationary performance. A plethora of works suggest that complex signals can be viewed as compositions of simpler subsystems (Qi et al., 2004; Young, 2011), motivating the use of decomposition-based forecasting (Wu et al., 2023). This calls for models that go beyond statistical extrapolation to uncover the intrinsic logic of evolving time-frequency patterns.

A prevailing trend in recent SOTAs is to improve performance in non-stationary LTSF by decomposing inputs into components with distinct *temporal* or *spectral* properties. Temporal-domain methods often separate trend and seasonal terms (Zhou et al., 2025a; Wu et al., 2021), linear and nonlinear patterns (Yu et al., 2025), or low- and high-frequency signals (Huang et al., 2025). While these approaches achieve empirical gains, they are largely heuristic and fail to capture the generative mechanisms driving non-stationary dynamics. This limitation is particularly pronounced in real-world, highly nonlinear systems, where prediction errors grow rapidly due to evolving spectrotemporal content (Lorenz, 1963; Osinga, 2018). A complementary line of work focuses on theoretically motivated decompositions that aim to extract interpretable components in transformed domains. For instance, DeRiTS (Fan et al., 2024) models multi-derivative stationary-frequency patterns. Its reliance on the global Fourier basis, however, hinders its ability to capture time-localized and aperiodic events. SimpleTM (Chen et al., 2025a) and WaveTS (Zhou et al., 2025b) address this by leveraging wavelet to disentangle trends and oscillatory components. Current works also exploit *inter-series dependen-*

*cies* (Wang et al., 2025; Yu et al., 2025), due to their critical role in strongly coupled systems where non-stationary behavior emerges from variable interactions.

In short, decomposition-based methods enhance long-term forecasting by breaking down series into simpler components, often aiming to "stationarize" them. However, this assumption is limiting, as real-world processes are rarely stationary, and forcing stationarization may suppress time-varying dynamics. Frequency-domain approaches (Chen et al., 2025a; Huang et al., 2025) reduce over-stationarization but struggle to adapt to dynamic changes due to rigid frameworks. These issues highlight the need for a framework that natively models non-stationarity at all levels.

Building on these insights, we propose Dual-Frac, a neural forecasting framework that uses nonlinear, time-varying neural fractional operators (NFOs) to model non-stationary time series, avoiding decomposition and stationarization while approximating the true generative process. Crucially, DualFrac's design ensures that every intermediate component remains non-stationary, thereby avoiding the information loss and over-smoothing common to classical decomposition-based methods. Each block combines static (data-independent) and

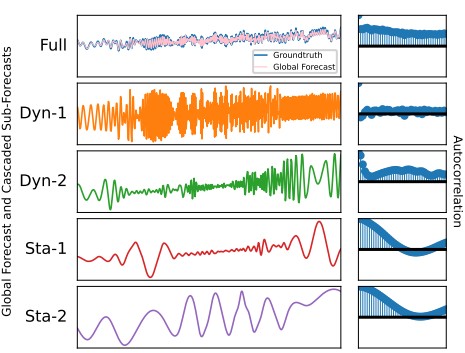

Figure 1: DualFrac's interleaved cascaded forecasts on a synthetic non-stationary signal: global forecast (pink) versus ground truth (blue), two dynamic sub-forecasts, capturing intermittent oscillations; and two static sub-forecasts, capturing more broad trends. Each component is provably non-stationary via learnable differential transforms; sample autocorrelations (right) highlight distinct quasi-periodic and intermittent dynamics.

dynamic (input-adaptive) NFOs, with Inter-NFOs capturing cross-variate dependencies and Intra-NFOs modeling temporal evolution, their outputs gated and merged. Cascaded layers integrate residuals, summing intermediate forecasts for robust extrapolation under distributional shifts. As shown in Fig. 1, DualFrac generates interpretable sub-forecasts, with dynamic NFOs capturing oscillatory patterns and static NFOs extracting trends, forming a global forecast. Notably, it also produces distinctly separated sub-forecasts that collectively align closely with the ground truth in the time-frequency domain (Fig. 2b). Experiments on diverse benchmarks show DualFrac's superior performance and expressivity. Our contributions are:

- We introduce DualFrac, a novel neural fractional cascading forecasting framework that fundamentally addresses long-term non-stationarity by leveraging a fractional time-frequency domain perspective. DualFrac adaptively models both intra- and inter-variate information while preserving diverse temporal patterns.

- We design a two-stage neural operator, comprising static and dynamic modules, to capture both stable and time-varying dynamics over sequences. This cascaded structure enables DualFrac to perform generalizable forecasting through provably non-stationary signal modeling.

- We validate DualFrac through extensive long-term forecasting experiments and thorough theoretical and empirical analyses, consistently outperforming SOTA baselines.

## 2 RELATED WORK

**Non-stationary Time Series.** Prior works apply stationarization as a preprocessing step, such as RevIN (Kim et al., 2021) and DAN (Liu et al., 2023c) for perform instance- or statistic-level normalization with learnable mappings between input/output or across variates. Recent works such as LiNo (Yu et al., 2025) and TwinsFormer (Zhou et al., 2025a) goes further by alternating between decomposed series to disentangle distinct dynamics, with spatial dependency and decomposition-based de-stationarization. DeRiTS (Fan et al., 2024) WaveTS (Zhou et al., 2025b) further improves by stationarizing on the frequency domain, operating with global dependency. However, most of these approaches rely on either residual heuristics or stationarization in fixed frequency bands. In

contrast, our method introduces a fractional-domain decomposition framework, theoretically supported by the view that any non-stationary process can be obtained from a family of sub-series, each learned in its own distribution, thus alleviating aforementioned issues.

**Learning on Transformed Domain.** Recent advances in time series forecasting have increasingly turned to transformed domain, such as Fourier and wavelet (Yi et al., 2025). Motivated by classical spectral decomposition techniques, many have incorporated domain conversion to enhance temporal representation. Depending on how such information is handled, existing approaches can be broadly grouped. Some models operate entirely in the time domain without any transformation (Zhou et al., 2021a; Liu et al., 2024a; Wang et al., 2024a), while others apply unified processing to all frequency components without distinguishing their roles Yi et al. (2024b); Zhou et al. (2022c). Some focus exclusively on low-frequency signals under the assumption that they carry the most predictive power Zhou et al. (2022a); Xu et al. (2024). More recent studies adopt weighted strategies that multicomponent transforms Zhou et al. (2022b); Zhang et al. (2024); Yi et al. (2024c). While prior methods offer promising results, they often isolate frequency components and rely on rigid spectral or basis assumptions, limiting their ability to model dynamical behaviors.

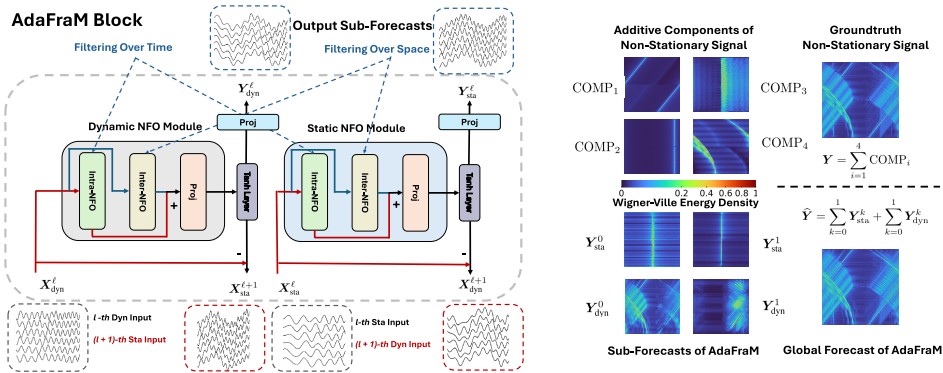

Figure 2: DualFrac's block structure and non-stationary decomposition on a synthetic test signal. **(a)** A single DualFrac block: the input is split into static and dynamic streams, each processed by Inter- and Intra-series FNOs; their gated outputs yield sub-forecasts, while the residuals feed the next layer. **(b)** Top: four artificial additive components ($\text{COMP}_1$–$\text{COMP}_4$) and their Wigner-Ville energy densities, summing to form a highly non-stationary ground-truth signal. Bottom: DualFrac's four learned sub-forecasts, each exhibiting strong non-stationary time-frequency characteristics. The final global forecast $\widehat{Y}$, formed by summing these sub-forecasts, closely matches the Wigner–Ville distribution of the ground truth $Y$, demonstrating accurate recovery of complex dynamics.

## 3 PRELIMINARIES

**Problem Formulation.** Formally, let the input be a multivariate series $\boldsymbol{X} \in \mathbb{R}^{C \times T}$, where $C$ denotes the number of variables (channels) and $T$ is the total number of time steps. At any given time step $t$, the forecasting model takes as input a lookback segment of length $L$, denoted by $\boldsymbol{X}_{t-L:t} = \{\boldsymbol{x}_{t-L}, \ldots, \boldsymbol{x}_t\}$, where each $\boldsymbol{x}_t \in \mathbb{R}^C$. The forecasting task aims to predict the next $F$ future steps: $\hat{\boldsymbol{Y}}_t = \{\hat{\boldsymbol{x}}_{t+1}, \ldots, \hat{\boldsymbol{x}}_{t+F}\} \in \mathbb{R}^{C \times F}$, where the ground truth sequence is denoted by $\boldsymbol{Y}_t = \{\boldsymbol{x}_{t+1}, \ldots, \boldsymbol{x}_{t+F}\}$. The forecasting model $\mathcal{F}(\cdot)$ learns a mapping from past observations to future predictions $\hat{\boldsymbol{Y}}_t = \mathcal{F}(\boldsymbol{X}_{t-L:t})$.

**Fractional Domain.** We leverage the *Fractional Fourier Transform* (FrFT) (Namias, 1980; Yu et al., 2023; Chen et al., 2025b), a classical linear transform to build our neural network that transforms temporal signals into a rich continuum of intermediate representations between the time and frequency domains. Unlike conventional transforms, signals purely into frequency or multiscale bases, the FrFT is a linear time–frequency operator parameterized by a rotation angle $\theta$ (also denoted $\alpha$). Formally, the FrFT can be interpreted as a rotation in the time–frequency plane in the

sense of Wigner–Ville, providing a continuum between the identity ($\theta = 0$), the Fourier transform ($\theta = \pi/2$), and the time reversal ($\theta = \pi$). Given a real-valued signal $X(t)$, $t$ is the time or space coordinate, the Wigner–Ville distribution is

$$W_X(t, \omega) = \int_{\mathbb{R}} X\left(t + \frac{\tau}{2}\right) X^*\left(t - \frac{\tau}{2}\right) e^{-j 2\pi \omega \tau} \, \mathrm{d}\tau, \tag{1}$$

which provides a quadratic time (space)–frequency representation. The action of the FrFT corresponds to a rotation of $W_X$ by angle $\theta$ in the $(t, \omega)$–plane, mapping content onto an intermediate *fractional domain*. Equivalently, one can view the FrFT as the projection

$$\mathcal{F}^\theta[X](\xi) = \iint_{\mathbb{R}^2} W_X(t, \omega) \, \delta\big(\xi - t\cos\theta - \omega\sin\theta\big) \, \mathrm{d}t \, \mathrm{d}\omega, \tag{2}$$

where $\delta(\cdot)$ is the Dirac distribution. We refer to $\xi$ as the *fractional domain coordinate*. Learning in this domain enables dynamic, data-adaptive time–frequency selectivity, which underpins DualFrac. The FrFT is defined as

**Definition 1** (FrFT). Let $X \in L^2(\mathbb{R})$. For each $\theta \in \mathbb{R}$, the fractional Fourier transform of order $\theta$ is

$$\mathcal{X}_\theta(\xi) \triangleq \mathcal{F}^\theta[X](\xi) = \int_{\mathbb{R}} X(t) K_\theta(\xi, t) \, dt \tag{3}$$

where the kernel $K_\theta(\xi, t)$ is defined as:

$$K_\theta(\xi, t) = \begin{cases} A_\theta \exp\left[\frac{j}{2}(\xi^2 + t^2)\cot\theta - j\xi t \csc\theta\right], \theta \notin \pi\mathbb{Z} \\ \delta(\xi - t), \quad \theta = 2k\pi \\ \delta(\xi + t), \quad \theta = (2k - 1)\pi \end{cases} \tag{4}$$

where $A_\theta = \sqrt{\frac{1 - j\cot\theta}{2\pi}}$ and $k \in \mathbb{Z}$. The inversion is $\mathcal{F}^{-\theta}$.

# 4 METHODOLOGY

## 4.1 FRACTIONAL NEURAL OPERATOR

Real-world time series often exhibit nonstationary behaviors, such as drifting instantaneous frequencies, intermittent structures, and amplitude-phase modulation, which are prevalent in chaotic or nonlinear systems. Traditional techniques like differencing or normalization attempt to stationarize the data but fail to address the geometric misalignment of correlation and energy in the time-frequency plane, where nonstationary signals often display distorted Wigner-Ville distribution, which deviate from the axis-aligned assumptions of fixed transform. To capture these, we employ fractional pseudo-differential operators (Prasad & Kumar, 2016; Upadhyay et al., 2013), which generalize convolution and differentiation with time (space)-frequency adaptivity.

**Definition 2** (Fractional Pseudo-Differential Operator). Let $a(t, \xi)$ be a sufficiently smooth time–frequency function (a *symbol*) on $\mathbb{R}_t \times \mathbb{R}_\xi$. For any $\theta \notin \pi\mathbb{Z}$, define the associated *fractional pseudo-differential operator*

$$T_a^\theta : L^2(\mathbb{R}) \to L^2(\mathbb{R}) \tag{3}$$

by

$$(T_a^\theta \phi)(t) = \int_{\mathbb{R}} K_{-\theta}(t, \xi) \, a(t, \xi) \, \widehat{\phi}^\theta(\xi) \mathrm{d}\xi, \tag{4}$$

where $\widehat{\phi}^\theta(\xi) = F^\theta[\phi](\xi)$ is the FrFT of angle $\theta$. In particular: If $a(t, \xi) = (i\,\xi\,\csc\theta)^m$, then $T_a^\theta$ reduces to the classical $m$-th order fractional derivative. If $a(t, \xi) = a(\xi)$ is independent of $x$, then $T_a^\theta$ is a fractional convolution operator.

To make this operator adaptive and learnable, we introduce the Neural Fractional Operator (NFO), a neural extension of $T_a^\theta$.

**Definition 3** (Neural Fractional Operator). Let $X \in \mathbb{R}^{L \times C}$ be a multivariate time series with $C$ channels, or let $\varphi \in L^2(\mathbb{R})^C$ be a multivariate signal in the continuous setting. The symbol $a(t, \xi)$ is factorized as:

$$a(t, \xi) = u(t) \, v(\xi), \tag{5}$$

where $u : \mathbb{R} \to \mathbb{R}^C$ and $v : \mathbb{R} \to \mathbb{R}^C$ are learned functions representing time-dependent and frequency-dependent profiles, respectively. In practice, $u(t)$ and $v(\xi)$ are predicted by hypernetworks $\text{HyperNet}^t$ and $\text{HyperNet}^\xi$, each instantiated as a one-layer MLP with Snake activation $A(x) = x + \sin^2(2x)/2$ (Belcák & Wattenhofer, 2022). The *Neural Fractional Operator* (NFO) $\mathcal{T}_a^\theta : L^2(\mathbb{R})^C \to L^2(\mathbb{R})^C$ of order $\theta \notin \pi\mathbb{Z}$ acts on a signal $\varphi \in L^2(\mathbb{R})^C$ as:

$$\boxed{\left(\mathcal{T}_a^\theta \varphi\right)(t) = u(t)\, \mathcal{F}^{-\theta}\left[v(\cdot) \odot \mathcal{F}^\theta[\varphi]\right](t)} \tag{6}$$

where $\mathcal{F}^\theta[\cdot]$ denotes the Fractional Fourier Transform of order $\theta$, and $\odot$ represents channel-wise multiplication.

NFO functions as a dynamic, signal-aware filter that operates jointly over time and frequency representations. When the NFO is independent of the input signal, its hypernetwork inputs are set to Fourier embeddings that depend solely on the length of the filtering axis, initialized with Fourier positional embeddings, and denoted as $\text{NFO}_{\text{sta}}$. Conversely, when the NFO depends on the input signal's content, its symbol factors are initialized based on the input, and it is denoted as $\text{NFO}_{\text{dyn}}$.

## 4.2 DualFrac Architecture

DualFrac is structured as a hierarchical decomposition architecture designed explicitly to handle the complex behaviors of non-stationary time series, as shown in Fig 2a. At its core, DualFrac consists of multiple stacked Neural Fractional Operator (NFO) modules arranged in a residual cascade, progressively extracting meaningful fractionally-aligned temporal and frequency-domain features. Each NFO block outputs two primary components: one is a decomposed intermediate component, projected by a linear layer to sub-forecast, contributing directly to the final forecast, and the other is a residual intermediate component carrying forward unresolved features for further decomposition by subsequent layers. To preserve overall trend within and among different series, we eliminate normalization layers and adopt an adaptive $\text{Tanh}$ activation function instead.

**Interleaved NFO Interaction.** Within each DualFrac block, the NFOs dynamically constructs fractional pseudo-differential operators using data-dependent and data-driven symbols, respectively. Concretely, given an input embedding $\boldsymbol{X} \in \mathbb{R}^{C \times L \times D}$, where $C$ denotes the number of variates (channels), $L$ the sequence length, and $D$ the embedding dimension, an NFO module performs fractional filtering along both temporal (intra-series) and variate (inter-series) dimensions independently, enabling the network to explicitly disentangle spatial and temporal dynamics. We consider two distinct symbol generation strategies: fractionally static and fractionally dynamic filtering.

**Fractionally Static NFO Module.** The static module generates symbols that capture global or slowly varying patterns, independent of instantaneous input features. For inter-series $\text{Inter-NFO}$ and intra-series $\text{Intra-NFO}$ NFOs, their outputs are fused to yield the statically filtered output with scaling parameter $\boldsymbol{\gamma}$,

$$\boldsymbol{Y}_{\text{sta}}^{\text{Inter}} = \text{Inter-NFO}_{\text{sta}}(\boldsymbol{X}), \tag{9a}$$

$$\boldsymbol{Y}_{\text{sta}}^{\text{Intra}} = \text{Intra-NFO}_{\text{sta}}(\boldsymbol{X}), \tag{9b}$$

$$\boldsymbol{Y}_{\text{sta}}^{\text{Out}} = \boldsymbol{\gamma} \tanh\left(\text{Linear}\left(\boldsymbol{Y}_{\text{sta}}^{\text{Inter}} + \boldsymbol{Y}_{\text{sta}}^{\text{Intra}}\right)\right) \tag{9c}$$

**Fractionally Dynamic NFO.** In contrast, the fractionally dynamic NFO employs an input-dependent, adaptive symbol parameterization to flexibly respond to instantaneous signal variations. Here, symbols are dynamically generated by single-layer Snake-activated copy of the input. Concretely,

$$\boldsymbol{Y}_{\text{dyn}}^{\text{Inter}} = \text{Inter-NFO}_{\text{dyn}}(\boldsymbol{X}), \tag{10a}$$

$$\boldsymbol{Y}_{\text{dyn}}^{\text{Intra}} = \text{Intra-NFO}_{\text{dyn}}(\boldsymbol{X}), \tag{10b}$$

$$\boldsymbol{Y}_{\text{dyn}}^{\text{Out}} = \boldsymbol{\gamma} \tanh\left(\text{Linear}\left(\boldsymbol{Y}_{\text{dyn}}^{\text{Inter}} + \boldsymbol{Y}_{\text{dyn}}^{\text{Intra}}\right)\right) \tag{10c}$$

**Cascaded Interweaving Forecasting.** To achieve effective multi-level forecasting, DualFrac implements a cascading residual structure. Each layer $\ell$ receives two distinct inputs: the static residual

component $\boldsymbol{X}_{\text{sta}}^{\ell}$ and dynamic residual component $\boldsymbol{X}_{\text{dyn}}^{\ell}$. Initially, these are obtained from the raw input $\boldsymbol{X}$ via separate linear embeddings:

$$\boldsymbol{X}_{\text{sta}}^{0} = \text{Linear}_{\text{sta}}(\boldsymbol{X}), \quad \boldsymbol{X}_{\text{dyn}}^{0} = \text{Linear}_{\text{dyn}}(\boldsymbol{X}). \tag{7}$$

And they experience filtering in a dual-stream pathway:

$$\begin{aligned} \boldsymbol{Y}_{\text{sta}}^{\text{Out},\ell} &= \text{StaticFiltering}(\boldsymbol{X}_{\text{sta}}^{\ell}), \\ \boldsymbol{Y}_{\text{dyn}}^{\text{Out},\ell} &= \text{DynamicFiltering}(\boldsymbol{X}_{\text{dyn}}^{\ell}). \end{aligned} \tag{8}$$

In DualFrac's interwoven architecture, the static and dynamic streams are not processed in isolation. Instead, the output of the static filtering ($\boldsymbol{Y}_{\text{sta}}^{\text{Out},\ell}$) may contribute to the dynamic input ($\boldsymbol{X}_{\text{dyn}}^{\ell+1}$) of the next layer, and vice versa. This interwoven exchange enables the model to capture complementary information by iteratively permuting the information flow between streams. These sub-forecasts refine and progressively adding well-aligned, non-stationary structures from the synthesized representations. The refined representations are forwarded as:

$$\boldsymbol{X}_{\text{dyn}}^{\ell+1} = \boldsymbol{X}_{\text{sta}}^{\ell} - \boldsymbol{Y}_{\text{sta}}^{\text{Out},\ell}, \tag{13a}$$

$$\boldsymbol{X}_{\text{sta}}^{\ell+1} = \boldsymbol{X}_{\text{dyn}}^{\ell} - \boldsymbol{Y}_{\text{dyn}}^{\text{Out},\ell}, \tag{13b}$$

$$\boldsymbol{Y}^{\ell} = \text{Proj}_{\text{sta}}\boldsymbol{Y}_{\text{sta}}^{\text{Out},\ell} + \text{Proj}_{\text{dyn}}\boldsymbol{Y}_{\text{dyn}}^{\text{Out},\ell} \tag{13c}$$

where $\text{Proj}_{\text{sta}}$ and $\text{Proj}_{\text{dyn}}$ are linear projection layers. The final forecast aggregates all sub-forecasts:

$$\hat{\boldsymbol{Y}} = \sum_{\ell=0}^{L-1} \boldsymbol{Y}^{\ell} \tag{14}$$

This cascading design enables DualFrac to decompose the input into distinct components while iteratively refining forecasts, resulting in more accurate long-term behaviors.

### 4.3 THEORETICAL ANALYSIS

We aim to show that the sum of outputs from a finite-depth neural architecture, where each layer performs a learnable fractional pseudo-differential transformation, can approximate a broad class of non-stationary processes exhibiting local time–frequency structures.

**Theorem 4** (Neural Fractional Approximation of Non-Stationary Processes). *Let $Y(t)$ be a non-stationary stochastic process exhibiting local regularities, as described in Dahlhaus (1996). For any $\varepsilon > 0$, there exist $m \geq 0$, and second-moment processes $\{X_i(t)\}_i^M \subset L^2(\Omega; H^{s+m,\theta})$, where $H^{s,\theta}$ is the $\theta$-Soboleve space (Prasad & Kumar, 2016), and $M$ learnable fractional pseudo-differential operators $\{T_{a_i}^{(\theta_i)}\}_{i=1}^M$, where each symbol $a_i(x,\xi)$ exhibits at most polynomial growth in $\xi$, and all angles satisfy $|\sin\theta_i| > 0$, such that each operator $T_{a_i}$ satisfies $\|T_{a_i}^{\theta_i}\phi\|_{H^{s,\theta}} \leq C\|\phi\|_{H^{s+m,\theta}}$ for a uniform constant $C$. The neural forecast is defined as $\widehat{Y}(t) := \sum_{i=1}^M T_{a_i}^{(\theta_i)} X_i(t)$, with the mean-squared error bound $\mathbb{E}\left[\left|Y(t) - \widehat{Y}(t)\right|^2\right] < \varepsilon$.*

Theorem 6 implies a spectral corollary: if each $Y_k(t)$ is energy-localized in frequency, then the Wigner–Ville time-frequency (or space-frequency) representation of $\widehat{Y}(t)$ satisfies the following property:

$$\lim_{M\to\infty} \left\| W_Y(t,\omega) - W_{\widehat{Y}_M}(t,\omega) \right\|_{L^1} = 0, \tag{9}$$

which provides a convergence guarantee when applied to our overall cascaded framework, stated as follows:

**Theorem 5** (Convergence of Cascaded NFO Decomposition). *Let $Y(t)$ be a stochastic process satisfying the conditions of Theorem 6. Define the residual sequence:*

$$R_0(t) := Y(t), \tag{10}$$

$$R_{k+1}(t) := R_k(t) - T_{a_k}^{(\theta_k)}[R_k](t), \quad k = 0, 1, \dots \tag{11}$$

324
325 *where each $T_{a_k}^{(\theta_k)}$ is a learnable neural fractional pseudo-differential operator. Then, for any $\varepsilon > 0$, there exists $N \in \mathbb{N}$ such that:*

$$\left\| \sum_{i=1}^{N} T_{a_i}^{(\theta_i)}[R_{i-1}](t) - Y(t) \right\|_{L^2} < \varepsilon \qquad (21)$$

## 5 EXPERIMENTS

### 5.1 EXPERIMENTAL SETUP

**Data and Baselines.** We evaluate DualFrac on 9 real-world benchmarks, following Wang et al. (2025); Zhou et al. (2025a), as well as a synthetic hyperchaotic datasets, WCN. We compare ours with classical and recent SOTAs, including Autoformer (Wu et al., 2021), CFPT (Kou et al., 2025), Crossformer (Zhang & Yan, 2023), DeRiTS (Fan et al., 2024), DLinear (Zeng et al., 2023), FED-former (Zhou et al., 2022c), FITS (Xu et al., 2023), FreTS (Yi et al., 2024a), Informer (Zhou et al., 2021b), iTransformer (Liu et al., 2024b), LiNo (Yu et al., 2025), PatchTST (Nie et al., 2022), SCINet (Liu et al., 2022a), SimpleTM (Chen et al., 2025a), Stationary (Liu et al., 2023a), Tex-Filter (Yi et al., 2024b), TiDE (Das et al., 2023), TimeKAN (Huang et al., 2025), TimeMixer (Wang et al., 2024b), TimeMixer++(Wang et al., 2025), TimesNet(Wu et al., 2023), Twinsformer (Zhou et al., 2025a) (Zhou et al., 2025b), WPMixer (Murad et al., 2025).

**Implementation.** All experiments are implemented in PyTorch 2.5.0 on 4 NVIDIA A100 (40GB) GPUs. Datasets and train/validation/test split are set up in accordance with those in works such as (Wang et al., 2025; 2024b). We report MSE and MAE as evaluation metrics.

Table 1: Average performance results of long-term time series forecasting. We set the lookback length as 96 and the prediction length in $\{96, 192, 336, 720\}$. The **best**, second and *third* results are highlighted. Full results are included in App. A.

| Models | Weather | | Solar | | ECL | | Traffic | | Exchange | | ETTh1 | | ETTh2 | | ETTm1 | | ETTm2 | | WCN | |
|---|---|---|---|---|---|---|---|---|---|---|---|---|---|---|---|---|---|---|---|---|
| Metrics | MSE | MAE | MSE | MAE | MSE | MAE | MSE | MAE | MSE | MAE | MSE | MAE | MSE | MAE | MSE | MAE | MSE | MAE | MSE | MAE |
| **DualFrac** | **.228** | **.254** | **.182** | **.241** | **.154** | **.248** | **.401** | .266 | **.346** | **.394** | **.398** | **.412** | *.338* | .382 | **.354** | **.369** | .247 | **.310** | **.646** | **.588** |
| CFPT | .240 | .267 | .291 | .336 | *.164* | .259 | .470 | .289 | .390 | .412 | .432 | .429 | .364 | .393 | .374 | .393 | .269 | *.315* | .726 | .630 |
| Twinsformer | .246 | .271 | .227 | .254 | .167 | .261 | **.406** | *.274* | .346 | .395 | .446 | .440 | .372 | .400 | .393 | .403 | .277 | .323 | .694 | .624 |
| LiNo | .241 | .270 | .275 | .320 | .164 | .260 | .465 | .295 | *.350* | .398 | .429 | .428 | .377 | .400 | .389 | .400 | .275 | .320 | .716 | .637 |
| TimeMixer++ | **.226** | .262 | .203 | *.258* | .165 | .253 | .416 | **.264** | .357 | .409 | .418 | .432 | .339 | **.380** | .368 | *.378* | .269 | .320 | .659 | .611 |
| TimeMixer | .240 | .272 | *.216* | .280 | .182 | .273 | .485 | .297 | .391 | .453 | .447 | .440 | .365 | .395 | .381 | .396 | .275 | .323 | .691 | .642 |
| iTransformer | .258 | .278 | .233 | .262 | .178 | .270 | .428 | .282 | .360 | .403 | .454 | .467 | .383 | .406 | .410 | .410 | .288 | .332 | .721 | .633 |
| PatchTST | .265 | .285 | .287 | .333 | .216 | .318 | .529 | .341 | .366 | .404 | .507 | .472 | .391 | .411 | .402 | .406 | .290 | .334 | .759 | .671 |
| Crossformer | .264 | .320 | .406 | .442 | .244 | .334 | .667 | .426 | .940 | .707 | .529 | .522 | .942 | .683 | .513 | .495 | .757 | .611 | 1.286 | .924 |
| TiDE | .270 | .320 | .347 | .417 | .252 | .344 | .760 | .473 | .370 | .413 | .541 | .507 | .611 | .550 | .419 | .419 | .358 | .404 | .962 | .832 |
| TimesNet | .259 | .286 | .402 | .374 | .193 | .303 | .620 | .336 | .416 | .443 | .458 | .450 | .414 | .427 | .400 | .406 | .291 | .333 | .951 | .836 |
| DLinear | .265 | .315 | .330 | .401 | .225 | .319 | .625 | .383 | .354 | .414 | .461 | .458 | .563 | .519 | .404 | .408 | .354 | .402 | .838 | .758 |
| SCINet | .292 | .363 | .282 | .375 | .268 | .365 | .804 | .509 | .750 | .626 | .747 | .647 | .954 | .723 | .485 | .481 | .571 | .537 | .863 | .772 |
| FEDformer | .309 | .360 | .328 | .383 | .213 | .327 | .609 | .376 | .519 | .429 | .498 | .484 | .436 | .449 | .448 | .452 | .304 | .349 | 1.274 | 1.104 |
| Stationary | .288 | .314 | .350 | .390 | .193 | .296 | .624 | .340 | .570 | .536 | .570 | .516 | .481 | .456 | .448 | .452 | .306 | .347 | .876 | .780 |
| Autoformer | .338 | .382 | .593 | .557 | .227 | .364 | .628 | .379 | .613 | .539 | .496 | .487 | .450 | .459 | .588 | .517 | .327 | .371 | 1.027 | .902 |
| WaveTS | .237 | .278 | .235 | .259 | .160 | *.253* | *.408* | .278 | .361 | .402 | .410 | .423 | **.332** | *.383* | .356 | .376 | **.244** | .311 | *.664* | *.614* |
| FITS | *.230* | .266 | .236 | .258 | .172 | .266 | .428 | .291 | .458 | .457 | *.412* | .427 | .337 | .385 | *.361* | .381 | .252 | .315 | .706 | .636 |
| DeRiTS | .293 | .321 | .361 | .340 | .293 | .376 | .976 | .545 | .427 | .505 | .682 | .566 | .435 | .439 | .715 | .555 | .321 | .359 | .967 | .849 |
| WPMixer | .235 | .283 | .250 | .263 | .175 | .264 | .448 | .316 | .426 | .471 | .418 | *.427* | .354 | .388 | .365 | .383 | .264 | .317 | .710 | .623 |
| TexFilter | .245 | .272 | .317 | .339 | .172 | .268 | .462 | .310 | .388 | .421 | .441 | .439 | .383 | .407 | .391 | .401 | .285 | .328 | .747 | .656 |
| SimpleTM | .243 | .271 | .184 | .247 | .166 | .260 | .444 | .289 | .371 | .412 | .422 | .428 | .353 | .391 | .381 | .396 | .275 | .322 | .676 | .625 |
| TimeKAN | .242 | .271 | .242 | .265 | .197 | .286 | .415 | .284 | .371 | .411 | .417 | .427 | .383 | .404 | .376 | .395 | .277 | .322 | .672 | .622 |
| FreTS | .288 | .314 | .350 | .390 | .193 | .296 | .624 | .340 | .461 | .454 | .570 | .537 | .526 | .516 | .481 | .456 | .306 | .347 | .884 | .777 |

### 5.2 MAIN RESULTS

**Comparative Study.** As shown in Tab. 1, DualFrac achieves comparable or superior results across a broad spectrum of datasets. It achieves 16 first-place and 3 second-place rankings out of 20 positions across two metrics on 10 datasets. Statistical analysis reveals a significant difference ($p < 0.005$) compared to TimeMixer++ and WaveTS, two best-performing baselines. DualFrac adaptively capture nonlinearly varying modes, this yields not only performance gains but also improves generalization. In particular, it demonstrates superior performance on datasets characterized by strong aperiodicity, chaotic patterns, or non-stationarity, such as ETTh1, Exchange, and WCN;

Table 2: Ablation and operator replacement studies for DualFrac. The symbol $\Delta$ represents the percentage of relative performance degradation. The symbols ✓ and ✗ indicate the presence or absence of a component, respectively. The following abbreviations are used: Sta (Static NFO), Dyn (Dynamic NFO), IL (Interleaved Architecture), DS (Dual-Stream Architecture), CR (Cascaded Residual), $\xi$ (fractional domain symbol factor), and $t$ (time domain symbol factor). The term "Operator" refers to the NFO or its replacement.

| Cases | Operator | Sta | Dyn | IL | DS | CR | $\xi$ | $t$ | IV | ECL | | ETTh1 | | Weather | | WCN | | $\Delta$ (%) | |
|---|---|---|---|---|---|---|---|---|---|---|---|---|---|---|---|---|---|---|---|
| | | | | | | | | | | MSE | MAE | MSE | MAE | MSE | MAE | MSE | MAE | MSE | MAE |
| Default | NFO | ✓ | ✓ | ✓ | ✓ | ✓ | ✓ | ✓ | ✓ | .154 | .248 | .398 | .412 | .228 | .254 | .646 | .588 | — | — |
| (1) | NFO | ✗ | ✓ | ✓ | ✓ | ✓ | ✓ | ✓ | ✓ | .183 | .282 | .464 | .481 | .271 | .286 | .764 | .685 | 18.14 | 14.89 |
| (2) | NFO | ✓ | ✗ | ✓ | ✓ | ✓ | ✓ | ✓ | ✓ | .172 | .276 | .468 | .467 | .264 | .285 | .733 | .670 | 14.64 | 12.70 |
| (3) | NFO | ✓ | ✓ | ✗ | ✓ | ✓ | ✓ | ✓ | ✓ | .170 | .283 | .446 | .466 | .250 | .287 | .717 | .674 | 1.77 | 13.71 |
| (4) | NFO | ✓ | ✓ | ✓ | ✗ | ✓ | ✓ | ✓ | ✓ | .174 | .280 | .419 | .473 | .245 | .284 | .682 | .660 | 7.83 | 12.94 |
| (5) | NFO | ✓ | ✓ | ✓ | ✓ | ✗ | ✓ | ✓ | ✓ | .168 | .269 | .425 | .444 | .244 | .276 | .696 | .624 | 7.66 | 7.76 |
| (6) | NFO | ✓ | ✓ | ✓ | ✓ | ✓ | ✗ | ✓ | ✓ | .165 | .276 | .409 | .445 | .238 | .275 | .682 | .652 | 4.97 | 9.61 |
| (7) | NFO | ✓ | ✓ | ✓ | ✓ | ✓ | ✓ | ✗ | ✓ | .171 | .263 | .418 | .455 | .245 | .284 | .682 | .652 | 7.28 | 9.80 |
| (8) | NFO | ✓ | ✓ | ✓ | ✓ | ✓ | ✓ | ✓ | ✗ | .167 | .265 | .437 | .458 | .250 | .286 | .712 | .673 | 9.53 | 11.27 |
| (9) | Fourier | ✓ | ✓ | ✓ | ✓ | ✓ | ✓ | ✓ | ✓ | .166 | .270 | .447 | .454 | .247 | .286 | .723 | .684 | 1.09 | 12.00 |
| (10) | FreMLP | ✓ | ✓ | ✓ | ✓ | ✓ | ✓ | ✓ | ✓ | .188 | .300 | .467 | .475 | .272 | .297 | .756 | .690 | 18.94 | 17.64 |
| (11) | AFNO | ✓ | ✓ | ✓ | ✓ | ✓ | ✓ | ✓ | ✓ | .186 | .306 | .487 | .496 | .284 | .313 | .784 | .703 | 22.27 | 21.64 |
| (12) | DeepFrFT | ✓ | ✓ | ✓ | ✓ | ✓ | ✓ | ✓ | ✓ | .179 | .300 | .476 | .480 | .266 | .295 | .740 | .673 | 16.76 | 16.77 |

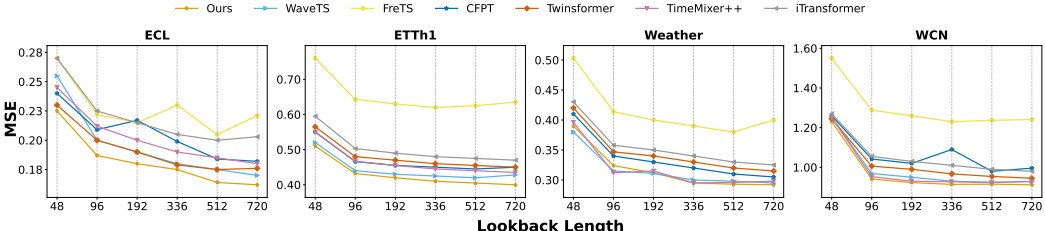

Figure 3: Influence of lookback horizon. We set lookback length in $\{48, 96, 192, 336, 512, 720\}$ and prediction length as 720. Increasing the lookback generally boosts DualFrac in a stable manner.

and maintains comparable or better long-term performance even on datasets with low forecastability, like Solar. These datasets pose significant challenges, as their non-stationarity cannot be effectively addressed through normalization or traditional seasonal-trend decomposition without losing critical information. When facing strong spatial coupling, DualFrac adeptly uncovers the nuanced interactions spanning multiple variates as shown in WCN and ECL.

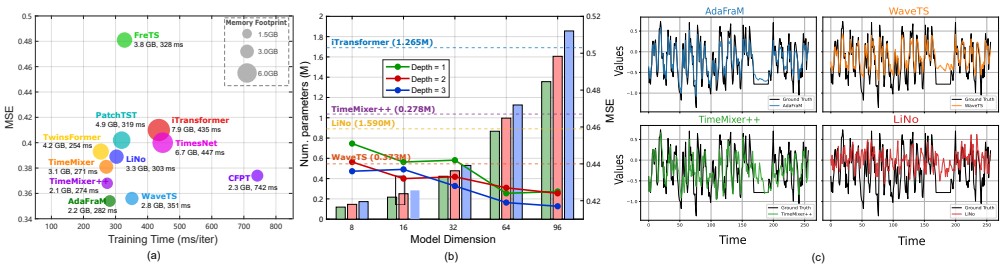

Figure 4: Efficiency, scalability, and case study of DualFrac on ETTh1. (a) Trade-off between MSE, training time, and memory footprint across models. Bubble size denotes memory usage. (b) Parameter efficiency and performance scaling with model dimension and depth. Bars show parameter count; lines show MSE. (c) Forecasting case study comparing DualFrac, WaveTS, TimeMixer++, and LiNo. DualFrac better captures sharp transitions and localized sharp shifts.

**Ablation Study.** To elucidate DualFrac's design, we conduct a comprehensive ablation study, systematically removing or modifying key modules, with results summarized in Tab. 2. The ablation variants are: (1) remove the static NFO, which captures data-independent symbol parameterization, to isolate its contribution; (2) exclude the dynamic NFO, responsible for input-adaptive symbol kernels, to assess its role; (3) replace the interleaved architecture, which processes intra- and inter-series axes jointly, with independent filtering to test its integrative benefit; (4) substitute the dual-stream architecture, combining parallel static and dynamic processing, with sequential operator stacking to evaluate parallel processing; (5) remove cascaded residual connections, enabling progressive refinement across layers, to rely solely on the final layer's output; (6) exclude the fractional coordinate $\xi$ from symbol parameterization to examine its role in frequency-based modeling; (7) exclude the original time/space coordinate $t$ from symbol parameterization to evaluate its contribution to spatial-temporal modeling; (8) replace inter-series NFO, which models cross-series dependencies, with intra-series NFO to assess its impact; (9) set the NFO's $\alpha$ to $\pi/2$ to mimic a Fourier operator (Yi et al., 2024a); (10) adopt the FreMLP operator (Yi et al., 2024a); (11) employ the Adaptive Fourier Neural Operator (AFNO) (Guibas et al., 2022); (12) use the DeepFrFT layer (Zhou et al., 2023). These modifications generally lead to performance degradation, with operator replacements showing the most significant declines.

**Lookback Analysis.** Increasing the lookback window is expected to boost forecasting by leveraging richer historical context. However, excessively increasing lookback length may increase noise or dilute critical features. To investigate this trade-off, we evaluate the impact of varying lookbacks on performance. Fig. 3 illustrates that DualFrac effectively utilizes extended lookback windows, exhibiting a positive correlation between input length and lower MSE in most cases. These underscore DualFrac's capability of capture long-term correlations, robust to distribution shifts.

## 5.3 MODEL ANALYSIS

**Necessity of Non-Stationarity.** To show whether off-the-shelf methods help DualFrac, we plug in RevIN (Kim et al., 2021), SAN (Liu et al., 2023b), and Dish-TS (Fan et al., 2023) then retrain the model. Intriguingly, as shown in Fig. 5, they overall brings negative gains. This supports our rationale: by learning mode decomposition, DualFrac leverages, rather than suppresses, the intrinsic time-frequency variability. Forcing stationarity can obscure predictive structure.

**Efficiency, Scaling Analysis and Show Cases.**
We compare DualFrac with the best-performing baselines in terms of MSE, memory footprint, and training speed on ETTh1. As shown in Fig. 4a, DualFrac not only achieves superior forecasting precision but also reduces memory consumption while maintaining high learning efficiency. The scaling behavior of DualFrac with respect to the model dimension $d_{\mathrm{model}}$ and depth $L$ is illustrated in Fig. 4b. We observe that DualFrac exhibits consistent performance gains as model parameters scale, but the most notable advantage occurs in the early stages. DualFrac quickly reaches a regime of competitive perfor-

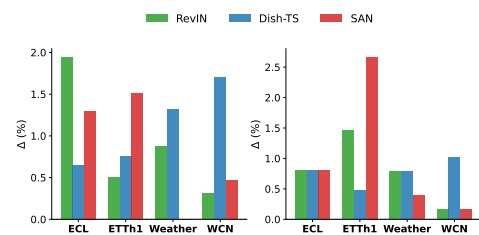

Figure 5: Average relative degradation of normalization methods.

mance with relatively fewer parameters, as indicated by the dashed lines. This demonstrates its effectiveness in achieving high performance without relying on excessive parameters. Fig. 4c showcases DualFrac's strength on ETTh1, capturing future variations amid quasi-periodic and intermittent dynamics.

## 6 CONCLUSION

We introduce DualFrac, a novel neural operator framework for non-stationary time series forecasting using fractional time-frequency representations to model inter- and intra-series dependencies. Its cascaded approach improves long-term accuracy, with experiments, ablation studies, and analyses across datasets confirming superiority over SOTA baselines, backed by strong theoretical foundations.

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

# A THEORETICAL ANALYSIS AND PROOF DETAILS

## A.1 PROOF OF NEURAL FRACTIONAL APPROXIMATION OF NON-STATIONARY PROCESSES

**Theorem 6.** *Let $Y(t)$ be a non-stationary stochastic process exhibiting local regularities, as described in Dahlhaus (1996). For any $\varepsilon > 0$, there exist $m \geq 0$, and second-moment processes $\{X_i(t)\}_i^M \subset L^2(\Omega; H^{s+m,\theta})$ and $M$ learnable fractional pseudo-differential operators $\{T_{a_i}^{(\theta_i)}\}_{i=1}^M$, where each symbol $a_i(x,\xi)$ exhibits at most polynomial growth in $\xi$, and all angles satisfy $|\sin\theta_i| > 0$, such that each operator $T_{a_i}$ satisfies $\|T_{a_i}^{\theta_i}\phi\|_{H^{s,\theta}} \leq C\|\phi\|_{H^{s+m,\theta}}$ for a uniform constant $C$. The neural forecast is defined as $\widehat{Y}(t) := \sum_{i=1}^M T_{a_i}^{(\theta_i)} X_i(t)$, with the mean-squared error bound $\mathbb{E}\left[\left|Y(t) - \widehat{Y}(t)\right|^2\right] < \varepsilon$.*

*Proof.* We work on the probability space $(\Omega, \mathcal{F}, \mathbb{P})$. The Sobolev norm associated with the fractional-Fourier angle $\theta$ is denoted $\|\cdot\|_{H^{s,\theta}}$, and the Hilbert space is defined as $\mathcal{H} := L^2(\Omega; H^{s,\theta}(\mathbb{R}))$ with norm

$$\|Z\|_{\mathcal{H}}^2 = \mathbb{E}\|Z\|_{H^{s,\theta}}^2. \tag{12}$$

We fix the angle $\theta$ and omit it from notation when unambiguous.

By the definition of local regularity (Dahlhaus, 1996), there exists a family of weakly stationary processes $\{Y_t(u) : u \in [0,1]\}$ such that, for some $\alpha > 0$,

$$\mathbb{E}|Y(t) - Y_t(u)|^2 = O(T^{-2\alpha}) \tag{13}$$

$$\text{whenever } |u - t/T| \leq cT^{-\gamma} \quad (0 < \gamma < 1). \tag{14}$$

Divide the index set $\{1, \ldots, T\}$ into $M$ disjoint blocks of equal length $b := \lfloor T/M \rfloor$, with block centers

$$u_j := (jb - 1/2)/T \quad (j = 1, \ldots, M).$$

Define the blockwise stationary surrogate

$$Y^{\text{LS}}(t) := Y_t(u_{j(t)}), \quad j(t) = \lceil t/b \rceil.$$

Using equation 14 and the fact that $\sum_{t=1}^T \mathbf{1}_{\{|u_{j(t)} - t/T| \leq cT^{-\gamma}\}} = T$, we obtain

$$\mathbb{E}\|Y - Y^{\text{LS}}\|_{\mathcal{H}}^2 = O(MT^{-2\alpha}). \tag{15}$$

Set $M := \lceil T^{2\alpha} \rceil$. Then equation 15 yields

$$\mathbb{E}\|Y - Y^{\text{LS}}\|_{\mathcal{H}}^2 < \varepsilon/2 \quad \text{for } T \text{ sufficiently large}. \tag{16}$$

Fix a block index $j$. Inside the block, $Y^{\text{LS}}$ coincides with the weakly stationary process $Y_t(u_j)$, whose fractional Fourier spectrum is square-integrable, so $Y_t(u_j) \in \mathcal{H}$.

The space $\mathcal{V}$ is defined as

$$\mathcal{V} = \overline{\mathrm{span}}\left\{ T_a^{(\vartheta)}[X] : a \in S_{m,\vartheta}^{\rho,\sigma}, \vartheta \in (0, 2\pi), X \in \mathcal{H} \right\} \subset \mathcal{H}.$$

Since symbols in $S_{m,\vartheta}^{\rho,\sigma}$ form a Sjöstrand/Wiener algebra and the corresponding fractional PDOs map the Feichtinger algebra $M^{1,1}$ into itself, the frame density theorem (Prasad & Kumar, 2016) implies

$$\mathcal{V} = \mathcal{H}. \tag{17}$$

By equation 17, for each block center $u_j$, there exist a second-moment process $X_j \in \mathcal{H}$, an angle $\theta_j \in (0, 2\pi)$ with $|\sin\theta_j| \geq \delta > 0$, and a symbol $a_j(x, \xi) \in S_{m,\theta_j}^{\rho,\sigma}$ with polynomial growth in $\xi$, such that the operator $T_{a_j}^{(\theta_j)}$ satisfies

$$\|T_{a_j}^{(\theta_j)}\phi\|_{H^{s,\theta}} \leq C\|\phi\|_{H^{s+m,\theta}}$$

and

$$\mathbb{E}\|Y_t(u_j) - T_{a_j}^{(\theta_j)}[X_j]\|_{H^{s,\theta}}^2 < \frac{\varepsilon}{2M}. \tag{18}$$

Define the global approximation

$$\widehat{Y}(t) = \sum_{j=1}^{M} T_{a_j}^{(\theta_j)}[X_j](t).$$

By the triangle inequality and orthogonality of the blocks,

$$\mathbb{E}\|Y - \widehat{Y}\|_{\mathcal{H}}^2 \leq 2\mathbb{E}\|Y - Y^{\mathrm{LS}}\|_{\mathcal{H}}^2 + 2\mathbb{E}\|Y^{\mathrm{LS}} - \widehat{Y}\|_{\mathcal{H}}^2$$

$$< \varepsilon + 2\sum_{j=1}^{M} \mathbb{E}\|Y_t(u_j) - T_{a_j}^{(\theta_j)}[X_j]\|_{H^{s,\theta}}^2$$

$$< \varepsilon + 2M \cdot \frac{\varepsilon}{2M} = \varepsilon.$$

Thus, the finite collection $\{X_j, \theta_j, a_j\}_{j=1}^{M}$ achieves the mean-square error bound $\varepsilon$.

$\square$

**Corollary 7** (Wigner–Ville Convergence of NFO Approximation). *Under the hypotheses of Theorem 6, further assume that each component $T_{a_i}^{(\theta_i)}X_i$ has its energy essentially confined to a disjoint frequency band. Then for the partial approximants*

$$\widehat{Y}_M(t) = \sum_{i=1}^{M} T_{a_i}^{(\theta_i)}[X_i](t)$$

*the corresponding Wigner–Ville distributions satisfy*

$$\lim_{M \to \infty} \left\| W_Y(t, \omega) - W_{\widehat{Y}_M}(t, \omega) \right\|_{L^1(\mathbb{R}^2)} = 0. \tag{19}$$

*Proof.* Let $e = f - g$ and $s = f + g$. The Moyal identity gives

$$\iint_{\mathbb{R}^2} |W_f - W_g|(t, \omega)\, dt\, d\omega \leq \|e\|_{L^2(\mathbb{R})} \|s\|_{L^2(\mathbb{R})}.$$

Apply this with $f = Y$ and $g = \widehat{Y}_M$. Since Theorem 6 guarantees $\|Y - \widehat{Y}_M\|_{L^2} \to 0$, and both $\|Y\|_{L^2}$ and $\|\widehat{Y}_M\|_{L^2}$ remain uniformly bounded, it follows that $\|W_Y - W_{\widehat{Y}_M}\|_{L^1} \to 0$. To handle cross-terms arising from the finite sum $\widehat{Y}_M = \sum_{i=1}^{M} f_i$, one uses the disjoint-band assumption: each $f_i$ has negligible Wigner overlap with $f_j$ when $i \neq j$. Concretely, if $\mathrm{supp}_\omega f_i \cap \mathrm{supp}_\omega f_j = \emptyset$, then $\iint |W_{f_i, f_j}|$ vanishes. Summing over $i \neq j$ therefore does not affect the limit.

Combining these two observations yields equation 19. $\square$

## A.2 Proof of Convergence of Cascaded NFO Decomposition

**Theorem 8.** *Let $Y(t)$ be a stochastic process satisfying the conditions of Theorem 6. Define the residual sequence:*

$$R_0(t) := Y(t), \tag{20}$$

$$R_{k+1}(t) := R_k(t) - T_{a_k}^{(\theta_k)}[R_k](t), \quad k = 0, 1, \dots \tag{21}$$

*where each $T_{a_k}^{(\theta_k)}$ is a learnable neural fractional pseudo-differential operator. Then, for any $\varepsilon > 0$, there exists $N \in \mathbb{N}$ such that:*

$$\left\| \sum_{i=1}^{N} T_{a_i}^{(\theta_i)}[R_{i-1}](t) - Y(t) \right\|_{L^2} < \varepsilon \tag{21}$$

*Proof.* Fix an arbitrary $\varepsilon > 0$. Choose a sequence of tolerances $\{\delta_k\}_{k=1}^{N} \subset (0, \infty)$ such that

$$\sum_{k=1}^{N} \delta_k \; < \; \varepsilon. \tag{22}$$

We will construct the residuals $R_k$ and operators $T_{a_k}^{(\theta_k)}$ by induction so that $\mathbb{E}\|R_k\|_{H^{s,\theta}}^2 < \sum_{i=1}^{k} \delta_i$.

Set $R_0(t) = Y(t)$. By Theorem 6, applied with tolerance $\delta_1$, there exists a process $X_1 \in L^2(\Omega; H^{s+m,\theta})$, an angle $\theta_1$, and a symbol $a_1 \in S_{m,\theta_1}^{\rho,\sigma}$ such that the corresponding operator $T_{a_1}^{(\theta_1)}$ satisfies

$$\mathbb{E}\left\| R_0 - T_{a_1}^{(\theta_1)}[X_1] \right\|_{H^{s,\theta}}^2 < \delta_1.$$

We then define

$$R_1 \; := \; R_0 \; - \; T_{a_1}^{(\theta_1)}[X_1], \tag{23}$$

so that

$$\mathbb{E}\left\| R_1 \right\|_{H^{s,\theta}}^2 < \delta_1. \tag{24}$$

Suppose for some $k \geq 1$ we have constructed $R_k$ satisfying $\mathbb{E}\|R_k\|_{H^{s,\theta}}^2 < \sum_{i=1}^{k} \delta_i$. Apply Theorem 6 to $R_k$ with tolerance $\delta_{k+1}$: there exist $X_{k+1}$, angle $\theta_{k+1}$, and symbol $a_{k+1}$ so that

$$\mathbb{E}\left\| R_k - T_{a_{k+1}}^{(\theta_{k+1})}[X_{k+1}] \right\|_{H^{s,\theta}}^2 < \delta_{k+1}.$$

Define

$$R_{k+1} := R_k - T_{a_{k+1}}^{(\theta_{k+1})}[X_{k+1}].$$

Then by the above inequality,

$$\mathbb{E}\left\| R_{k+1} \right\|_{H^{s,\theta}}^2 < \delta_{k+1} \quad \Longrightarrow \quad \mathbb{E}\left\| R_{k+1} \right\|_{H^{s,\theta}}^2 < \sum_{i=1}^{k+1} \delta_i. \tag{25}$$

This completes the induction. After $N$ steps, we have

$$\sum_{i=1}^{N} T_{a_i}^{(\theta_i)}[X_i](t) \; = \; Y(t) \; - \; R_N(t),$$

hence

$$\mathbb{E}\left\| \sum_{i=1}^{N} T_{a_i}^{(\theta_i)}[X_i] \; - \; Y \right\|_{H^{s,\theta}}^2 = \mathbb{E}\left\| R_N \right\|_{H^{s,\theta}}^2 < \sum_{i=1}^{N} \delta_i \overset{equation\ 22}{<} \varepsilon.$$

Since $X_i = R_{i-1}$ was the choice in each step, this completes the proof of Theorem 8. $\square$

Table 3: Full results for the long-term forecasting task. The lookback window size is set to 96, with prediction lengths of 96, 192, 336, and 720. Avg represents the average results across all four prediction lengths.

| Models | | WaveTS | | FITS | | DeRiTS | | WPMixer | | TexFilter | | FreTS | | SimpleTM | | TimeKAN | |
|---|---|---|---|---|---|---|---|---|---|---|---|---|---|---|---|---|---|
| Metrics | | MSE | MAE | MSE | MAE | MSE | MAE | MSE | MAE | MSE | MAE | MSE | MAE | MSE | MAE | MSE | MAE |
| Weather | 96 | 0.167 | 0.223 | 0.145 | 0.199 | 0.216 | 0.270 | 0.168 | 0.205 | 0.162 | 0.207 | 0.173 | 0.223 | 0.162 | 0.207 | 0.162 | 0.208 |
| | 192 | 0.210 | 0.258 | 0.190 | 0.243 | 0.264 | 0.304 | 0.209 | 0.270 | 0.210 | 0.250 | 0.245 | 0.285 | 0.208 | 0.248 | 0.207 | 0.249 |
| | 336 | 0.256 | 0.294 | 0.238 | 0.282 | 0.312 | 0.335 | 0.263 | 0.289 | 0.263 | 0.290 | 0.321 | 0.338 | 0.263 | 0.290 | 0.263 | 0.290 |
| | 720 | 0.315 | 0.336 | 0.310 | 0.332 | 0.380 | 0.375 | 0.339 | 0.339 | 0.339 | 0.340 | 0.414 | 0.410 | 0.340 | 0.341 | 0.338 | 0.340 |
| | Avg | 0.237 | 0.278 | 0.230 | 0.266 | 0.293 | 0.321 | 0.235 | 0.283 | 0.235 | 0.272 | 0.288 | 0.314 | 0.243 | 0.271 | 0.242 | 0.271 |
| Solar | 96 | 0.180 | 0.252 | 0.178 | 0.245 | 0.304 | 0.349 | 0.192 | 0.248 | 0.202 | 0.252 | 0.321 | 0.380 | 0.163 | 0.232 | 0.187 | 0.258 |
| | 192 | 0.221 | 0.235 | 0.219 | 0.234 | 0.350 | 0.315 | 0.232 | 0.246 | 0.325 | 0.361 | 0.346 | 0.369 | 0.182 | 0.247 | 0.228 | 0.241 |
| | 336 | 0.238 | 0.266 | 0.238 | 0.266 | 0.355 | 0.344 | 0.252 | 0.268 | 0.343 | 0.371 | 0.357 | 0.387 | 0.193 | 0.257 | 0.245 | 0.272 |
| | 720 | 0.302 | 0.282 | 0.310 | 0.286 | 0.434 | 0.354 | 0.325 | 0.290 | 0.397 | 0.373 | 0.375 | 0.424 | 0.199 | 0.252 | 0.309 | 0.289 |
| | Avg | 0.235 | 0.259 | 0.236 | 0.258 | 0.361 | 0.340 | 0.250 | 0.263 | 0.317 | 0.339 | 0.350 | 0.390 | 0.184 | 0.247 | 0.242 | 0.265 |
| ECL | 96 | 0.131 | 0.227 | 0.145 | 0.242 | 0.275 | 0.362 | 0.148 | 0.240 | 0.147 | 0.245 | 0.169 | 0.273 | 0.141 | 0.235 | 0.174 | 0.266 |
| | 192 | 0.146 | 0.240 | 0.157 | 0.252 | 0.277 | 0.364 | 0.161 | 0.250 | 0.160 | 0.251 | 0.182 | 0.286 | 0.151 | 0.247 | 0.182 | 0.273 |
| | 336 | 0.162 | 0.256 | 0.174 | 0.269 | 0.291 | 0.376 | 0.177 | 0.265 | 0.173 | 0.267 | 0.200 | 0.304 | 0.173 | 0.267 | 0.197 | 0.286 |
| | 720 | 0.200 | 0.288 | 0.213 | 0.301 | 0.329 | 0.402 | 0.215 | 0.302 | 0.210 | 0.309 | 0.222 | 0.321 | 0.201 | 0.293 | 0.236 | 0.320 |
| | Avg | 0.160 | 0.253 | 0.172 | 0.266 | 0.293 | 0.376 | 0.175 | 0.264 | 0.172 | 0.268 | 0.193 | 0.296 | 0.166 | 0.260 | 0.197 | 0.286 |
| Traffic | 96 | 0.382 | 0.266 | 0.401 | 0.280 | 0.961 | 0.542 | 0.431 | 0.312 | 0.430 | 0.294 | 0.612 | 0.338 | 0.410 | 0.274 | 0.389 | 0.272 |
| | 192 | 0.394 | 0.270 | 0.415 | 0.286 | 0.973 | 0.547 | 0.411 | 0.307 | 0.452 | 0.307 | 0.613 | 0.340 | 0.430 | 0.280 | 0.401 | 0.276 |
| | 336 | 0.409 | 0.278 | 0.429 | 0.290 | 0.959 | 0.536 | 0.443 | 0.311 | 0.470 | 0.316 | 0.618 | 0.328 | 0.449 | 0.290 | 0.416 | 0.284 |
| | 720 | 0.447 | 0.298 | 0.468 | 0.308 | 1.010 | 0.556 | 0.505 | 0.329 | 0.498 | 0.323 | 0.653 | 0.355 | 0.486 | 0.309 | 0.455 | 0.305 |
| | Avg | 0.408 | 0.278 | 0.428 | 0.291 | 0.976 | 0.545 | 0.448 | 0.316 | 0.462 | 0.310 | 0.624 | 0.340 | 0.444 | 0.289 | 0.415 | 0.284 |
| Exchange | 96 | 0.086 | 0.204 | 0.109 | 0.235 | 0.143 | 0.255 | 0.102 | 0.220 | 0.091 | 0.211 | 0.111 | 0.237 | 0.092 | 0.212 | 0.094 | 0.213 |
| | 192 | 0.177 | 0.300 | 0.229 | 0.350 | 0.240 | 0.355 | 0.202 | 0.310 | 0.186 | 0.305 | 0.219 | 0.335 | 0.185 | 0.308 | 0.183 | 0.309 |
| | 336 | 0.322 | 0.411 | 0.400 | 0.463 | 0.387 | 0.456 | 0.360 | 0.433 | 0.380 | 0.449 | 0.421 | 0.476 | 0.335 | 0.422 | 0.331 | 0.420 |
| | 720 | 0.860 | 0.693 | 1.0995 | 0.781 | 0.940 | 0.938 | 1.041 | 0.923 | 0.896 | 0.712 | 1.092 | 0.769 | 0.872 | 0.705 | 0.875 | 0.702 |
| | Avg | 0.361 | 0.402 | 0.458 | 0.457 | 0.427 | 0.505 | 0.426 | 0.471 | 0.388 | 0.421 | 0.461 | 0.454 | 0.371 | 0.412 | 0.371 | 0.411 |
| ETTh1 | 96 | 0.367 | 0.391 | 0.374 | 0.396 | 0.625 | 0.531 | 0.368 | 0.379 | 0.382 | 0.402 | 0.513 | 0.491 | 0.366 | 0.392 | 0.367 | 0.395 |
| | 192 | 0.404 | 0.414 | 0.407 | 0.416 | 0.665 | 0.550 | 0.419 | 0.419 | 0.430 | 0.429 | 0.534 | 0.504 | 0.422 | 0.421 | 0.414 | 0.420 |
| | 336 | 0.427 | 0.432 | 0.430 | 0.436 | 0.710 | 0.574 | 0.438 | 0.433 | 0.472 | 0.451 | 0.588 | 0.535 | 0.440 | 0.438 | 0.445 | 0.434 |
| | 720 | 0.440 | 0.455 | 0.435 | 0.458 | 0.730 | 0.608 | 0.446 | 0.460 | 0.481 | 0.473 | 0.643 | 0.616 | 0.463 | 0.462 | 0.444 | 0.459 |
| | Avg | 0.410 | 0.423 | 0.412 | 0.427 | 0.682 | 0.566 | 0.418 | 0.423 | 0.441 | 0.439 | 0.570 | 0.537 | 0.422 | 0.428 | 0.417 | 0.427 |
| ETTh2 | 96 | 0.267 | 0.333 | 0.273 | 0.339 | 0.380 | 0.400 | 0.281 | 0.336 | 0.293 | 0.343 | 0.476 | 0.458 | 0.281 | 0.338 | 0.290 | 0.340 |
| | 192 | 0.332 | 0.375 | 0.334 | 0.377 | 0.442 | 0.435 | 0.350 | 0.380 | 0.374 | 0.396 | 0.512 | 0.493 | 0.355 | 0.387 | 0.375 | 0.392 |
| | 336 | 0.349 | 0.396 | 0.356 | 0.398 | 0.465 | 0.461 | 0.374 | 0.405 | 0.417 | 0.430 | 0.552 | 0.551 | 0.365 | 0.401 | 0.423 | 0.435 |
| | 720 | 0.380 | 0.428 | 0.384 | 0.427 | 0.452 | 0.459 | 0.412 | 0.432 | 0.449 | 0.460 | 0.562 | 0.560 | 0.413 | 0.436 | 0.443 | 0.449 |
| | Avg | 0.332 | 0.383 | 0.337 | 0.385 | 0.435 | 0.439 | 0.354 | 0.388 | 0.383 | 0.407 | 0.526 | 0.516 | 0.353 | 0.391 | 0.383 | 0.404 |
| ETTm1 | 96 | 0.301 | 0.344 | 0.306 | 0.348 | 0.691 | 0.541 | 0.309 | 0.346 | 0.321 | 0.361 | 0.386 | 0.398 | 0.321 | 0.361 | 0.322 | 0.361 |
| | 192 | 0.338 | 0.365 | 0.340 | 0.369 | 0.708 | 0.550 | 0.350 | 0.369 | 0.367 | 0.387 | 0.459 | 0.444 | 0.360 | 0.380 | 0.357 | 0.383 |
| | 336 | 0.367 | 0.384 | 0.373 | 0.388 | 0.719 | 0.558 | 0.372 | 0.394 | 0.401 | 0.409 | 0.495 | 0.464 | 0.390 | 0.404 | 0.382 | 0.401 |
| | 720 | 0.416 | 0.412 | 0.424 | 0.419 | 0.742 | 0.572 | 0.430 | 0.422 | 0.477 | 0.448 | 0.585 | 0.516 | 0.454 | 0.438 | 0.445 | 0.435 |
| | Avg | 0.356 | 0.376 | 0.361 | 0.381 | 0.715 | 0.555 | 0.365 | 0.383 | 0.391 | 0.401 | 0.481 | 0.456 | 0.381 | 0.396 | 0.376 | 0.395 |
| ETTm2 | 96 | 0.162 | 0.252 | 0.165 | 0.256 | 0.227 | 0.308 | 0.170 | 0.254 | 0.175 | 0.258 | 0.192 | 0.274 | 0.173 | 0.257 | 0.174 | 0.255 |
| | 192 | 0.215 | 0.292 | 0.219 | 0.294 | 0.284 | 0.338 | 0.228 | 0.293 | 0.240 | 0.301 | 0.280 | 0.339 | 0.238 | 0.299 | 0.239 | 0.299 |
| | 336 | 0.263 | 0.326 | 0.271 | 0.328 | 0.339 | 0.370 | 0.290 | 0.330 | 0.311 | 0.347 | 0.334 | 0.361 | 0.296 | 0.338 | 0.301 | 0.340 |
| | 720 | 0.335 | 0.373 | 0.352 | 0.382 | 0.434 | 0.419 | 0.367 | 0.390 | 0.414 | 0.405 | 0.417 | 0.413 | 0.393 | 0.395 | 0.395 | 0.396 |
| | Avg | 0.244 | 0.311 | 0.252 | 0.315 | 0.321 | 0.359 | 0.264 | 0.317 | 0.285 | 0.328 | 0.306 | 0.347 | 0.275 | 0.322 | 0.277 | 0.322 |
| Wilson-Cowan | 96 | 0.264 | 0.330 | 0.279 | 0.347 | 0.385 | 0.462 | 0.283 | 0.339 | 0.297 | 0.357 | 0.352 | 0.422 | 0.272 | 0.341 | 0.271 | 0.339 |
| | 192 | 0.612 | 0.574 | 0.646 | 0.606 | 0.892 | 0.806 | 0.655 | 0.591 | 0.689 | 0.623 | 0.815 | 0.738 | 0.625 | 0.585 | 0.621 | 0.582 |
| | 336 | 0.812 | 0.743 | 0.856 | 0.736 | 1.182 | 0.992 | 0.868 | 0.728 | 0.913 | 0.767 | 1.081 | 0.908 | 0.825 | 0.755 | 0.821 | 0.751 |
| | 720 | 0.969 | 0.807 | 1.041 | 0.854 | 1.409 | 1.136 | 1.035 | 0.833 | 1.088 | 0.878 | 1.288 | 1.040 | 0.982 | 0.819 | 0.978 | 0.815 |
| | Avg | 0.664 | 0.614 | 0.706 | 0.636 | 0.967 | 0.849 | 0.710 | 0.623 | 0.747 | 0.656 | 0.884 | 0.777 | 0.676 | 0.625 | 0.672 | 0.622 |

Table 4: Full results for the long-term forecasting task. We set the lookback window size $L$ as 96 and the prediction length as $\tau \in \{96, 192, 336, 720\}$. Avg means the average results from all four prediction lengths.

| Models | | DualFrac | | CFPT | | Twinsformer | | LiNo | | TimeMixer++ | | TimeMixer | | iTransformer | | PatchTST | | Crossformer | | TiDE | | TimesNet | | DLinear | | SCINet | | FEDformer | | Nonstationary | | Autoformer | |
|---|---|---|---|---|---|---|---|---|---|---|---|---|---|---|---|---|---|---|---|---|---|---|---|---|---|---|---|---|---|---|---|---|---|---|---|
| Metrics | | MSE | MAE | MSE | MAE | MSE | MAE | MSE | MAE | MSE | MAE | MSE | MAE | MSE | MAE | MSE | MAE | MSE | MAE | MSE | MAE | MSE | MAE | MSE | MAE | MSE | MAE | MSE | MAE | MSE | MAE | MSE | MAE |
| Weather | 96 | **0.150** | **0.184** | 0.154 | 0.200 | 0.161 | 0.201 | 0.154 | 0.199 | 0.155 | 0.205 | 0.163 | 0.209 | 0.174 | 0.214 | 0.195 | 0.271 | 0.202 | 0.261 | 0.172 | 0.220 | 0.195 | 0.252 | 0.221 | 0.306 | 0.217 | 0.296 | 0.173 | 0.223 | 0.266 | 0.336 |
| | 192 | 0.212 | **0.240** | 0.203 | 0.242 | 0.211 | 0.248 | 0.205 | 0.248 | 0.208 | 0.250 | 0.221 | 0.254 | 0.221 | 0.254 | 0.209 | 0.277 | 0.242 | 0.298 | 0.219 | 0.261 | 0.261 | 0.340 | 0.276 | 0.336 | 0.245 | 0.285 | 0.221 | 0.338 | 0.307 | 0.367 |
| | 336 | **0.225** | **0.261** | 0.261 | 0.286 | 0.266 | 0.291 | 0.262 | 0.290 | 0.237 | 0.245 | 0.251 | 0.287 | 0.284 | 0.301 | 0.273 | 0.332 | 0.287 | 0.335 | 0.282 | 0.331 | 0.309 | 0.378 | 0.339 | 0.380 | 0.321 | 0.338 | 0.359 | 0.395 |
| | 720 | 0.324 | **0.331** | 0.340 | 0.339 | 0.347 | 0.343 | 0.343 | 0.342 | 0.312 | 0.334 | 0.339 | 0.341 | 0.358 | 0.347 | 0.356 | 0.349 | 0.351 | 0.386 | 0.345 | 0.382 | 0.377 | 0.427 | 0.403 | 0.428 | 0.414 | 0.410 | 0.419 | 0.428 |
| | Avg | **0.228** | **0.254** | 0.240 | 0.267 | 0.246 | 0.271 | 0.241 | 0.270 | 0.226 | 0.262 | 0.240 | 0.272 | 0.258 | 0.278 | 0.265 | 0.285 | 0.264 | 0.320 | 0.270 | 0.320 | 0.259 | 0.286 | 0.265 | 0.315 | 0.292 | 0.363 | 0.309 | 0.360 | 0.338 | 0.382 |
| Solar | 96 | **0.152** | **0.224** | 0.232 | 0.318 | 0.193 | **0.224** | 0.171 | 0.254 | 0.171 | 0.231 | 0.189 | 0.299 | 0.203 | 0.237 | 0.265 | 0.323 | 0.312 | 0.399 | 0.373 | 0.358 | 0.290 | 0.378 | 0.237 | 0.344 | 0.286 | 0.341 | 0.321 | 0.380 | 0.456 | 0.446 |
| | 192 | 0.181 | **0.237** | 0.265 | 0.247 | 0.223 | 0.250 | 0.237 | 0.298 | 0.218 | 0.263 | 0.222 | 0.283 | 0.233 | 0.261 | 0.288 | 0.332 | 0.371 | 0.410 | 0.339 | 0.416 | 0.397 | 0.376 | 0.320 | 0.398 | 0.280 | 0.380 | 0.291 | 0.337 | 0.346 | 0.369 | 0.588 | 0.561 |
| | 336 | 0.193 | **0.250** | 0.323 | 0.381 | 0.246 | 0.268 | 0.296 | 0.336 | 0.212 | 0.269 | 0.231 | 0.292 | 0.248 | 0.273 | 0.301 | 0.339 | 0.495 | 0.515 | 0.368 | 0.430 | 0.420 | 0.380 | 0.353 | 0.415 | 0.304 | 0.389 | 0.354 | 0.416 | 0.357 | 0.387 | 0.595 | 0.588 |
| | 720 | **0.202** | **0.255** | 0.342 | 0.397 | 0.245 | 0.272 | 0.395 | 0.393 | 0.212 | 0.270 | 0.223 | 0.285 | 0.249 | 0.275 | 0.305 | 0.336 | 0.526 | 0.542 | 0.370 | 0.425 | 0.381 | 0.375 | 0.424 | 0.733 | 0.633 |
| | Avg | **0.182** | **0.241** | 0.291 | 0.336 | 0.227 | 0.254 | 0.275 | 0.320 | 0.203 | 0.258 | 0.216 | 0.280 | 0.233 | 0.262 | 0.287 | 0.333 | 0.406 | 0.442 | 0.347 | 0.417 | 0.402 | 0.374 | 0.330 | 0.401 | 0.282 | 0.375 | 0.328 | 0.383 | 0.350 | 0.390 | 0.593 | 0.557 |
| ECL | 96 | **0.125** | **0.223** | 0.136 | 0.231 | 0.139 | 0.233 | 0.138 | 0.233 | 0.135 | 0.222 | 0.153 | 0.247 | 0.148 | 0.240 | 0.190 | 0.296 | 0.219 | 0.314 | 0.237 | 0.329 | 0.168 | 0.272 | 0.210 | 0.302 | 0.247 | 0.345 | 0.193 | 0.308 | 0.169 | 0.273 | 0.201 | 0.317 |
| | 192 | **0.140** | **0.232** | 0.153 | 0.246 | 0.158 | 0.252 | 0.155 | 0.250 | 0.147 | 0.235 | 0.166 | 0.256 | 0.162 | 0.253 | 0.199 | 0.304 | 0.231 | 0.322 | 0.236 | 0.330 | 0.184 | 0.322 | 0.210 | 0.305 | 0.257 | 0.355 | 0.201 | 0.315 | 0.182 | 0.286 | 0.222 | 0.334 |
| | 336 | **0.162** | **0.256** | 0.168 | 0.265 | 0.172 | 0.267 | 0.171 | 0.267 | 0.164 | 0.245 | 0.185 | 0.277 | 0.178 | 0.269 | 0.217 | 0.319 | 0.246 | 0.337 | 0.249 | 0.344 | 0.198 | 0.300 | 0.223 | 0.319 | 0.269 | 0.369 | 0.214 | 0.329 | 0.200 | 0.304 | 0.231 | 0.443 |
| | 720 | **0.187** | **0.279** | 0.199 | 0.293 | 0.200 | 0.293 | 0.191 | 0.290 | 0.212 | 0.310 | 0.225 | 0.310 | 0.225 | 0.317 | 0.258 | 0.352 | 0.280 | 0.363 | 0.284 | 0.373 | 0.220 | 0.320 | 0.258 | 0.350 | 0.299 | 0.390 | 0.246 | 0.355 | 0.222 | 0.321 | 0.254 | 0.361 |
| | Avg | **0.154** | **0.248** | 0.164 | 0.259 | 0.167 | 0.261 | 0.164 | 0.260 | 0.165 | 0.253 | 0.182 | 0.273 | 0.178 | 0.273 | 0.216 | 0.322 | 0.244 | 0.334 | 0.252 | 0.344 | 0.193 | 0.303 | 0.225 | 0.319 | 0.268 | 0.365 | 0.213 | 0.327 | 0.193 | 0.296 | 0.227 | 0.364 |
| Traffic | 96 | **0.375** | **0.254** | 0.444 | 0.274 | 0.382 | 0.260 | 0.429 | 0.276 | 0.392 | **0.253** | 0.462 | 0.285 | 0.395 | 0.268 | 0.526 | 0.347 | 0.644 | 0.429 | 0.805 | 0.493 | 0.593 | 0.321 | 0.650 | 0.396 | 0.788 | 0.499 | 0.587 | 0.366 | 0.612 | 0.338 | 0.613 | 0.388 |
| | 192 | **0.393** | **0.265** | 0.460 | 0.280 | 0.392 | 0.267 | 0.450 | 0.289 | 0.402 | 0.258 | 0.473 | 0.296 | 0.417 | 0.276 | 0.522 | 0.332 | 0.665 | 0.431 | 0.756 | 0.474 | 0.617 | 0.336 | 0.598 | 0.370 | 0.789 | 0.505 | 0.604 | 0.373 | 0.613 | 0.340 | 0.616 | 0.382 |
| | 336 | **0.396** | **0.259** | 0.477 | 0.289 | 0.410 | 0.276 | 0.468 | 0.297 | 0.428 | 0.263 | 0.498 | 0.296 | 0.433 | 0.283 | 0.517 | 0.334 | 0.674 | 0.420 | 0.762 | 0.477 | 0.629 | 0.336 | 0.605 | 0.373 | 0.797 | 0.508 | 0.621 | 0.383 | 0.618 | 0.328 | 0.622 | 0.337 |
| | 720 | **0.440** | **0.278** | 0.499 | 0.313 | 0.442 | 0.292 | 0.514 | 0.320 | 0.441 | 0.282 | 0.506 | 0.313 | 0.467 | 0.302 | 0.552 | 0.352 | 0.683 | 0.424 | 0.719 | 0.449 | 0.640 | 0.350 | 0.645 | 0.394 | 0.841 | 0.523 | 0.626 | 0.382 | 0.653 | 0.355 | 0.660 | 0.408 |
| | Avg | **0.401** | **0.264** | 0.470 | 0.289 | 0.406 | 0.274 | 0.465 | 0.295 | 0.416 | 0.264 | 0.485 | 0.297 | 0.428 | 0.282 | 0.529 | 0.341 | 0.667 | 0.426 | 0.760 | 0.473 | 0.620 | 0.336 | 0.625 | 0.383 | 0.804 | 0.509 | 0.609 | 0.376 | 0.624 | 0.340 | 0.628 | 0.379 |
| Exchange | 96 | 0.082 | **0.198** | 0.189 | 0.207 | **0.081** | **0.200** | 0.084 | 0.203 | 0.085 | 0.214 | 0.090 | 0.235 | 0.086 | 0.206 | 0.088 | 0.205 | 0.256 | 0.367 | 0.094 | 0.218 | 0.107 | 0.234 | 0.088 | 0.218 | 0.267 | 0.396 | 0.148 | 0.278 | 0.111 | 0.237 | 0.197 | 0.323 |
| | 192 | 0.174 | **0.293** | 0.210 | 0.312 | **0.172** | **0.295** | 0.176 | 0.298 | 0.175 | 0.313 | 0.187 | 0.343 | 0.177 | 0.299 | 0.176 | 0.299 | 0.470 | 0.509 | 0.184 | 0.307 | 0.226 | 0.344 | 0.176 | 0.315 | 0.351 | 0.459 | 0.271 | 0.315 | 0.219 | 0.335 | 0.300 | 0.369 |
| | 336 | **0.311** | **0.409** | 0.314 | 0.437 | 0.316 | 0.409 | 0.316 | 0.409 | 0.353 | 0.473 | 0.353 | 0.473 | 0.331 | 0.417 | **0.301** | **0.397** | 1.268 | 0.883 | 0.349 | 0.431 | 0.367 | 0.448 | 0.313 | 0.427 | 1.324 | 0.853 | 0.460 | 0.427 | 0.421 | 0.476 | 0.509 | 0.524 |
| | 720 | 0.816 | **0.674** | 0.846 | 0.692 | **0.812** | **0.677** | 0.823 | 0.682 | 0.851 | 0.689 | 0.934 | 0.761 | 0.847 | 0.691 | 0.901 | 0.714 | 1.767 | 1.068 | 0.852 | 0.698 | 0.964 | 0.746 | 0.839 | 0.695 | 1.058 | 0.797 | 1.195 | 0.695 | 1.092 | 0.769 | 1.447 | 0.941 |
| | Avg | 0.346 | **0.394** | 0.390 | 0.412 | 0.346 | **0.395** | 0.350 | 0.398 | 0.357 | 0.409 | 0.391 | 0.453 | 0.360 | 0.403 | 0.366 | 0.404 | 0.940 | 0.707 | 0.370 | 0.413 | 0.416 | 0.443 | 0.354 | 0.414 | 0.750 | 0.629 | 0.461 | 0.454 | 0.613 | 0.539 |
| ETTh1 | 96 | **0.354** | **0.382** | 0.372 | 0.391 | 0.385 | 0.401 | 0.378 | 0.395 | 0.361 | 0.403 | 0.375 | 0.400 | 0.386 | 0.405 | 0.460 | 0.447 | 0.423 | 0.448 | 0.479 | 0.402 | 0.384 | 0.402 | 0.397 | 0.412 | 0.654 | 0.599 | 0.395 | 0.424 | 0.513 | 0.491 | 0.449 | 0.459 |
| | 192 | **0.385** | **0.383** | 0.425 | 0.421 | 0.439 | 0.431 | 0.442 | 0.480 | 0.416 | 0.441 | 0.429 | 0.421 | 0.441 | 0.512 | 0.477 | 0.429 | 0.436 | 0.429 | 0.446 | 0.441 | 0.719 | 0.631 | 0.469 | 0.470 | 0.534 | 0.500 | 0.482 |
| | 336 | **0.419** | **0.431** | 0.467 | 0.442 | 0.480 | 0.452 | 0.455 | 0.438 | 0.430 | 0.434 | 0.484 | 0.458 | 0.487 | 0.458 | 0.546 | 0.496 | 0.570 | 0.546 | 0.565 | 0.515 | 0.491 | 0.469 | 0.489 | 0.467 | 0.778 | 0.659 | 0.530 | 0.499 | 0.588 | 0.535 | 0.521 | 0.496 |
| | 720 | **0.432** | **0.450** | 0.466 | 0.461 | 0.480 | 0.474 | 0.459 | 0.456 | 0.467 | 0.451 | 0.498 | 0.482 | 0.503 | 0.491 | 0.544 | 0.517 | 0.653 | 0.621 | 0.594 | 0.558 | 0.521 | 0.500 | 0.513 | 0.510 | 0.836 | 0.699 | 0.598 | 0.544 | 0.643 | 0.616 | 0.514 | 0.512 |
| | Avg | **0.398** | **0.412** | 0.432 | 0.429 | 0.446 | 0.440 | 0.429 | **0.428** | 0.418 | 0.432 | 0.447 | 0.454 | 0.467 | 0.507 | 0.472 | 0.529 | 0.522 | 0.541 | 0.507 | 0.458 | 0.459 | 0.498 | 0.484 | 0.570 | 0.576 | 0.494 | 0.498 | 0.487 |
| ETTh2 | 96 | **0.279** | **0.332** | 0.285 | 0.336 | 0.292 | 0.345 | 0.292 | 0.340 | **0.276** | **0.328** | 0.289 | 0.341 | 0.297 | 0.349 | 0.308 | 0.355 | 0.745 | 0.584 | 0.400 | 0.440 | 0.340 | 0.374 | 0.340 | 0.394 | 0.707 | 0.621 | 0.358 | 0.397 | 0.476 | 0.458 | 0.346 | 0.388 |
| | 192 | 0.356 | **0.384** | 0.363 | 0.388 | 0.375 | 0.395 | 0.375 | 0.391 | **0.342** | **0.379** | 0.372 | 0.392 | 0.380 | 0.400 | 0.405 | 0.877 | 0.656 | 0.528 | 0.509 | 0.402 | 0.414 | 0.430 | 0.479 | 0.860 | 0.689 | 0.429 | 0.439 | 0.512 | 0.493 | 0.456 | 0.452 |
| | 336 | 0.335 | **0.402** | 0.413 | 0.426 | 0.417 | 0.429 | 0.418 | 0.426 | **0.346** | **0.398** | 0.386 | 0.414 | 0.428 | 0.432 | 0.427 | 0.445 | 1.043 | 0.731 | 0.643 | 0.571 | 0.452 | 0.452 | 0.468 | 0.839 | 0.661 | 1.249 | 0.838 | 0.496 | 0.487 | 0.552 | 0.551 | 0.482 | 0.486 |
| | 720 | 0.382 | **0.410** | 0.390 | 0.422 | 0.406 | 0.430 | 0.422 | 0.441 | 0.392 | 0.415 | 0.412 | 0.434 | 0.427 | 0.445 | 0.436 | 0.450 | 1.104 | 0.763 | 0.874 | 0.679 | 0.462 | 0.468 | 0.839 | 0.661 | 1.249 | 0.838 | 0.463 | 0.474 | 0.562 | 0.560 | 0.515 | 0.511 |
| | Avg | 0.338 | **0.382** | 0.364 | 0.393 | 0.372 | 0.400 | 0.377 | 0.400 | **0.339** | **0.380** | 0.365 | 0.395 | 0.383 | 0.406 | 0.391 | 0.411 | 0.942 | 0.683 | 0.611 | 0.550 | 0.414 | 0.427 | 0.563 | 0.510 | 0.954 | 0.723 | 0.436 | 0.449 | 0.526 | 0.516 | 0.450 | 0.459 |
| ETTm1 | 96 | **0.288** | **0.320** | 0.316 | 0.356 | 0.325 | 0.364 | 0.322 | 0.361 | 0.310 | 0.334 | 0.320 | 0.357 | 0.334 | 0.368 | 0.352 | 0.374 | 0.404 | 0.426 | 0.364 | 0.387 | 0.338 | 0.375 | 0.346 | 0.374 | 0.418 | 0.438 | 0.379 | 0.419 | 0.386 | 0.398 | 0.505 | 0.475 |
| | 192 | **0.341** | **0.356** | 0.354 | 0.380 | 0.372 | 0.390 | 0.365 | 0.384 | 0.348 | 0.362 | 0.361 | 0.381 | 0.390 | 0.393 | 0.374 | 0.387 | 0.450 | 0.451 | 0.398 | 0.404 | 0.374 | 0.387 | 0.382 | 0.391 | 0.439 | 0.450 | 0.426 | 0.441 | 0.459 | 0.444 | 0.553 | 0.496 |
| | 336 | **0.366** | **0.378** | 0.383 | 0.400 | 0.406 | 0.412 | 0.401 | 0.408 | 0.376 | 0.391 | 0.390 | 0.404 | 0.426 | 0.420 | 0.421 | 0.414 | 0.532 | 0.515 | 0.428 | 0.425 | 0.410 | 0.411 | 0.415 | 0.415 | 0.490 | 0.485 | 0.445 | 0.459 | 0.495 | 0.464 | 0.621 | 0.537 |
| | 720 | **0.422** | **0.411** | 0.444 | 0.434 | 0.467 | 0.448 | 0.469 | 0.447 | 0.440 | 0.423 | 0.454 | 0.441 | 0.491 | 0.459 | 0.462 | 0.449 | 0.666 | 0.589 | 0.487 | 0.461 | 0.478 | 0.450 | 0.473 | 0.451 | 0.595 | 0.550 | 0.543 | 0.490 | 0.585 | 0.516 | 0.671 | 0.561 |
| | Avg | **0.354** | **0.369** | 0.374 | 0.393 | 0.393 | 0.403 | 0.389 | 0.404 | 0.368 | **0.378** | 0.381 | 0.396 | 0.410 | 0.410 | 0.402 | 0.406 | 0.513 | 0.495 | 0.419 | 0.419 | 0.400 | 0.406 | 0.404 | 0.408 | 0.485 | 0.481 | 0.448 | 0.452 | 0.481 | 0.456 | 0.588 | 0.517 |
| ETTm2 | 96 | **0.165** | **0.250** | 0.167 | 0.249 | 0.173 | 0.256 | 0.171 | 0.254 | **0.245** | 0.175 | 0.258 | 0.180 | 0.264 | 0.183 | 0.270 | 0.287 | 0.366 | 0.207 | 0.305 | 0.187 | 0.267 | 0.193 | 0.293 | 0.286 | 0.377 | 0.203 | 0.287 | 0.192 | 0.274 | 0.255 | 0.339 |
| | 192 | **0.209** | **0.287** | 0.232 | 0.292 | 0.239 | 0.300 | 0.237 | 0.298 | 0.229 | 0.291 | 0.237 | 0.299 | 0.250 | 0.309 | 0.255 | 0.314 | 0.414 | 0.492 | 0.290 | 0.364 | 0.249 | 0.309 | 0.284 | 0.361 | 0.399 | 0.445 | 0.269 | 0.328 | 0.280 | 0.339 | 0.281 | 0.340 |
| | 336 | **0.256** | **0.313** | 0.290 | 0.311 | 0.298 | 0.339 | 0.296 | 0.336 | 0.303 | 0.343 | 0.298 | 0.340 | 0.311 | 0.348 | 0.309 | 0.347 | 0.597 | 0.542 | 0.377 | 0.422 | 0.321 | 0.351 | 0.382 | 0.429 | 0.637 | 0.591 | 0.325 | 0.366 | 0.334 | 0.361 | 0.339 | 0.372 |
| | 720 | 0.359 | **0.388** | 0.385 | 0.389 | 0.397 | 0.397 | 0.395 | 0.393 | 0.373 | 0.399 | 0.391 | 0.396 | 0.412 | 0.407 | 0.412 | 0.404 | 1.730 | 1.042 | 0.558 | 0.524 | 0.408 | 0.403 | 0.558 | 0.523 | 0.421 | 0.415 | 0.417 | 0.413 | 0.433 | 0.432 |
| | Avg | **0.247** | **0.310** | 0.269 | 0.315 | 0.277 | 0.323 | 0.275 | 0.320 | 0.269 | 0.320 | 0.275 | 0.323 | 0.288 | 0.332 | 0.290 | 0.334 | 0.757 | 0.611 | 0.358 | 0.404 | 0.291 | 0.333 | 0.354 | 0.402 | 0.571 | 0.537 | 0.304 | 0.349 | 0.306 | 0.347 | 0.327 | 0.371 |
| WCN | 96 | **0.257** | **0.320** | 0.291 | 0.339 | 0.277 | 0.334 | 0.284 | 0.343 | **0.267** | **0.319** | 0.296 | 0.354 | 0.301 | 0.362 | 0.510 | 0.507 | 0.375 | 0.451 | 0.378 | 0.454 | 0.337 | 0.411 | 0.342 | 0.408 | 0.507 | 0.611 | 0.432 | 0.412 | 0.498 |
| | 192 | **0.595** | **0.558** | 0.672 | 0.597 | 0.636 | 0.598 | 0.656 | 0.668 | 0.657 | 0.569 | 0.602 | 0.606 | 0.665 | 0.602 | 0.770 | 0.705 | 1.189 | 0.874 | 0.887 | 0.799 | 0.892 | 0.802 | 0.378 | 0.705 | 0.791 | 0.741 | 1.163 | 1.059 | 0.808 | 0.732 | 0.949 | 0.843 |
| | 336 | **0.789** | **0.686** | 0.900 | 0.743 | 0.656 | 0.708 | 0.877 | 0.725 | **0.805** | 0.702 | 0.870 | 0.746 | 0.878 | 0.726 | 0.910 | 0.778 | 1.573 | 1.082 | 1.188 | 0.983 | 1.140 | 1.003 | 1.027 | 0.884 | 1.061 | 0.913 | 1.550 | 1.275 | 1.058 | 0.817 | 0.921 | 1.249 | 1.060 |
| | 720 | **0.941** | **0.787** | 1.042 | 0.839 | 1.006 | 0.858 | 1.049 | 0.872 | 0.953 | 0.854 | 0.947 | 0.864 | 1.055 | 0.851 | 1.124 | 0.905 | 1.871 | 1.235 | 1.398 | 1.094 | 1.393 | 1.087 | 1.230 | 1.031 | 1.257 | 1.028 | 1.876 | 1.470 | 1.292 | 1.034 | 1.504 | 1.206 |
| | Avg | **0.646** | **0.588** | 0.726 | 0.630 | 0.694 | 0.624 | 0.716 | 0.637 | 0.659 | 0.611 | 0.691 | 0.642 | 0.721 | 0.633 | 0.759 | 0.671 | 1.286 | 0.924 | 0.962 | 0.832 | 0.951 | 0.836 | 0.838 | 0.758 | 0.863 | 0.772 | 1.274 | 1.104 | 0.876 | 0.780 | 1.027 | 0.902 |

# B FULL RESULTS

## B.1 MAIN EXPERIMENTS

The full results of main comparison experiments are presented in Table 4 and 3.

## B.2 ABLATION STUDIES

The detailed results of ablation experiments are provided in Tab. 5.

Table 5: Ablation and operator replacement studies for DualFrac across multiple prediction lengths. ✓ and ✗ indicate the presence or removal of a component. (1): w/o Static NFO; (2): w/o Dynamic NFO; (3): w/o Interleaved Architecture; (4): w/o Dual-Stream Architecture; (5): w/o Cascaded Residual; (6): w/o $\xi$; (7): w/o $x$; (8): w/o inter-series spatial non-stationary; (9)-(12): Operator replacements (specific variants to be detailed as needed).

| Dataset | Horizon | Default | | (1) | | (2) | | (3) | | (4) | | (5) | | (6) | | (7) | | (8) | | (9) | | (10) | | (11) | | (12) | |
|---|---|---|---|---|---|---|---|---|---|---|---|---|---|---|---|---|---|---|---|---|---|---|---|---|---|---|---|---|---|
| | | MSE | MAE | MSE | MAE | MSE | MAE | MSE | MAE | MSE | MAE | MSE | MAE | MSE | MAE | MSE | MAE | MSE | MAE | MSE | MAE | MSE | MAE | MSE | MAE | MSE | MAE |
| ECL | 96 | .125 | .223 | .132 | .224 | .137 | .224 | .142 | .260 | .139 | .260 | .139 | .233 | .132 | .253 | .136 | .224 | .137 | .224 | .138 | .225 | .156 | .272 | .144 | .264 | .145 | .270 |
| | 192 | .140 | .232 | .171 | .270 | .154 | .253 | .153 | .274 | .160 | .258 | .156 | .257 | .148 | .250 | .152 | .257 | .146 | .258 | .175 | .286 | .172 | .292 | .160 | .280 |
| | 336 | .162 | .256 | .204 | .311 | .181 | .297 | .175 | .285 | .183 | .296 | .174 | .275 | .179 | .269 | .180 | .280 | .174 | .280 | .176 | .286 | .199 | .296 | .195 | .309 | .190 | .310 |
| | 720 | .187 | .279 | .227 | .323 | .215 | .331 | .210 | .312 | .213 | .307 | .205 | .312 | .201 | .320 | .210 | .300 | .204 | .300 | .205 | .310 | .222 | .346 | .235 | .358 | .220 | .340 |
| | Avg | .154 | .248 | .183 | .282 | .172 | .276 | .170 | .283 | .174 | .280 | .168 | .269 | .165 | .276 | .171 | .263 | .167 | .265 | .166 | .270 | .188 | .300 | .186 | .306 | .179 | .300 |
| ETTh1 | 96 | .354 | .382 | .390 | .410 | .395 | .410 | .417 | .430 | .417 | .430 | .370 | .440 | .378 | .422 | .370 | .406 | .370 | .420 | .398 | .421 | .415 | .441 | .435 | .459 | .420 | .450 |
| | 192 | .385 | .383 | .470 | .465 | .463 | .446 | .418 | .449 | .401 | .427 | .416 | .415 | .393 | .416 | .400 | .420 | .417 | .440 | .432 | .423 | .452 | .443 | .472 | .462 | .465 | .450 |
| | 336 | .419 | .431 | .481 | .514 | .498 | .509 | .482 | .491 | .444 | .496 | .444 | .453 | .451 | .466 | .400 | .480 | .470 | .480 | .472 | .475 | .491 | .498 | .512 | .520 | .500 | .510 |
| | 720 | .432 | .450 | .514 | .535 | .518 | .503 | .468 | .494 | .461 | .527 | .462 | .484 | .438 | .492 | .460 | .500 | .465 | .490 | .486 | .498 | .508 | .519 | .529 | .541 | .520 | .510 |
| | Avg | .398 | .412 | .464 | .481 | .468 | .467 | .446 | .466 | .419 | .473 | .425 | .444 | .409 | .455 | .418 | .455 | .447 | .454 | .467 | .475 | .487 | .496 | .476 | .480 |
| Weather | 96 | .150 | .184 | .171 | .195 | .167 | .204 | .152 | .195 | .159 | .216 | .161 | .192 | .159 | .200 | .159 | .216 | .152 | .190 | .153 | .198 | .188 | .225 | .194 | .226 | .170 | .220 |
| | 192 | .212 | .240 | .257 | .265 | .239 | .261 | .237 | .287 | .226 | .278 | .237 | .259 | .237 | .277 | .237 | .287 | .228 | .266 | .248 | .275 | .249 | .289 | .240 | .270 |
| | 336 | .225 | .261 | .279 | .295 | .267 | .311 | .253 | .293 | .235 | .287 | .232 | .285 | .233 | .284 | .235 | .287 | .253 | .293 | .250 | .298 | .272 | .297 | .283 | .323 | .270 | .320 |
| | 720 | .324 | .331 | .377 | .388 | .363 | .365 | .359 | .373 | .361 | .358 | .361 | .369 | .359 | .347 | .361 | .358 | .359 | .373 | .356 | .381 | .356 | .392 | .409 | .417 | .385 | .370 |
| | Avg | .228 | .254 | .271 | .286 | .264 | .285 | .250 | .287 | .245 | .284 | .244 | .276 | .238 | .275 | .245 | .284 | .250 | .286 | .247 | .286 | .272 | .297 | .284 | .313 | .266 | .295 |
| WCN | 96 | .257 | .320 | .280 | .360 | .286 | .379 | .280 | .360 | .269 | .370 | .360 | .269 | .333 | .278 | .339 | .280 | .367 | .280 | .367 | .281 | .368 | .315 | .370 | .323 | .377 | .300 | .380 |
| | 192 | .595 | .558 | .722 | .643 | .704 | .606 | .699 | .632 | .648 | .624 | .669 | .599 | .652 | .643 | .648 | .600 | .690 | .630 | .692 | .644 | .685 | .632 | .727 | .663 | .710 | .610 |
| | 336 | .789 | .686 | .944 | .779 | .876 | .789 | .870 | .776 | .823 | .782 | .824 | .744 | .823 | .742 | .823 | .780 | .870 | .775 | .853 | .807 | .933 | .808 | .975 | .817 | .880 | .790 |
| | 720 | .941 | .787 | 1.110 | .958 | 1.067 | .906 | 1.014 | .928 | .987 | .862 | 1.023 | .819 | .986 | .885 | .987 | .860 | 1.010 | .920 | 1.064 | .917 | 1.090 | .950 | 1.113 | .953 | 1.070 | .910 |
| | Avg | .646 | .588 | .764 | .685 | .733 | .670 | .717 | .674 | .682 | .660 | .696 | .624 | .682 | .652 | .682 | .652 | .712 | .673 | .723 | .684 | .756 | .690 | .784 | .703 | .740 | .673 |

# C EXPERIMENTAL DETAILS

We present details of datasets, evaluation metrics and experiments in this appendix.

Table 6: Effectiveness analysis of non-stationarity across multiple prediction lengths.

| Cases | | Default | | + RevIN | | + FISH-TS | | + SAN | |
|---|---|---|---|---|---|---|---|---|---|
| Metrics | | MSE | MAE | MSE | MAE | MSE | MAE | MSE | MAE |
| ECL | 96 | .125 | .223 | .126 | .223 | .126 | .224 | .128 | .229 |
| | 192 | .140 | .232 | .143 | .232 | .140 | .233 | .141 | .233 |
| | 336 | .162 | .256 | .166 | .260 | .164 | .258 | .164 | .256 |
| | 720 | .187 | .279 | .191 | .284 | .189 | .284 | .189 | .280 |
| | Avg | .154 | .248 | .157 | .250 | .155 | .250 | .156 | .250 |
| ETTh1 | 96 | .354 | .382 | .356 | .384 | .358 | .384 | .359 | .388 |
| | 192 | .385 | .383 | .387 | .387 | .389 | .387 | .391 | .393 |
| | 336 | .419 | .431 | .424 | .443 | .422 | .431 | .425 | .447 |
| | 720 | .432 | .450 | .434 | .458 | .436 | .454 | .439 | .458 |
| | Avg | .398 | .412 | .400 | .418 | .401 | .414 | .404 | .423 |
| Weather | 96 | .150 | .184 | .153 | .185 | .153 | .184 | .151 | .184 |
| | 192 | .212 | .240 | .214 | .241 | .214 | .240 | .211 | .240 |
| | 336 | .225 | .261 | .228 | .263 | .229 | .263 | .226 | .261 |
| | 720 | .324 | .331 | .325 | .333 | .326 | .335 | .325 | .335 |
| | Avg | .228 | .254 | .230 | .256 | .231 | .256 | .228 | .255 |
| WCN | 96 | .257 | .320 | .258 | .321 | .262 | .327 | .261 | .322 |
| | 192 | .595 | .558 | .596 | .559 | .616 | .569 | .595 | .556 |
| | 336 | .789 | .686 | .790 | .687 | .794 | .688 | .792 | .689 |
| | 720 | .941 | .787 | .942 | .788 | .953 | .793 | .943 | .789 |
| | Avg | .645 | .588 | .647 | .589 | .656 | .594 | .648 | .589 |

## C.1 DATASETS

We evaluate the performance of different models for long-term forecasting on 9 well-established long-term datasets, including Weather, Traffic, ECL, Exchange, Solar-Energy, and ETT datasets (ETTh1, ETTh2, ETTm1, ETTm2). Furthermore, we adopt Wilson-Cowan Network (WCN) (Wilson, 2019), which is a high-dimensional hyperchaotic dynamical system to evaluate the long-term forecasting performance on non-stationary complicated dynamics. We detail the descriptions of experimental data as follows:

- **ETT**: The ETT datasets, namely *ETTh1*, *ETTh2*, *ETTm1*, and *ETTm2*, consist of measurements from electrical transformers. ETTh datasets (ETTh1, ETTh2) record seven variables including voltage, current, and temperature on an hourly basis, while ETTm datasets (ETTm1, ETTm2) capture the same seven variables every 15 minutes, from July 2016 to July 2018.

- **ECL**: This dataset tracks the electricity consumption metrics of 321 clients, recorded every 15 minutes, reflecting both residential and industrial usage. It involves a large number of variables, with 321 distinctive measures of consumption patterns.

- **Exchange Rate**: Featuring daily records of exchange rates for eight major currencies, this dataset encompasses a time span from 1990 to 2016 and includes eight variables per timestamp, aiding in the analysis of long-term economic trends.

- **Traffic**: Capturing the dynamics of traffic flow and occupancy rates with 862 sensors, this dataset provides hourly data across various freeways in the San Francisco Bay Area from January 2015 to December 2016. The dataset is rich in dimensions, focusing on a broad range of traffic-related variables.

- **Weather**: This dataset is gathered every 10 minutes from the Max Planck Institute for Biogeochemistry's weather station and includes 21 comprehensive meteorological variables such as temperature, humidity, and wind speed throughout 2020. It offers a detailed look into climatic conditions with a high resolution in both time and variable space.

- **Solar**: This dataset records the power output of 137 photovoltaic plants in Alabama at 10-minute resolution during 2016. It provides multi-site solar production data, often used for multivariate forecasting benchmarks in renewable energy prediction tasks.

- **WCN**: A dataset or synthetic simulation suite derived from networks of coupled Wilson–Cowan oscillators arranged in chains, grids, or sparse arrays. When coupling inhibitory nodes to excitatory nodes between oscillators, the system exhibits *hyperchaotic dynamics*, quantified by multiple positive Lyapunov exponents that scale approximately linearly with the number of oscillators. The time series data comprise excitatory/inhibitory population activities across nodes under varying coupling strengths, enabling analysis of complex, high-dimensional chaos.

Table 7: Dataset Descriptions. The dataset size is organized as (Train, Validation, Test). Forecastability is computed based on predictability scores from (Liu et al., 2022b).

| Dataset | Dim | Series Length | Dataset Size | Forecastability |
|---------|-----|---------------|--------------|-----------------|
| ETTm1 | 7 | {96, 192, 336, 720} | (34465, 11521, 11521) | 0.46 |
| ETTm2 | 7 | {96, 192, 336, 720} | (34465, 11521, 11521) | 0.55 |
| ETTh1 | 7 | {96, 192, 336, 720} | (8545, 2881, 2881) | 0.38 |
| ETTh2 | 7 | {96, 192, 336, 720} | (8545, 2881, 2881) | 0.45 |
| ECL | 321 | {96, 192, 336, 720} | (18317, 2633, 5261) | 0.77 |
| Traffic | 862 | {96, 192, 336, 720} | (12185, 1757, 3509) | 0.68 |
| Weather | 21 | {96, 192, 336, 720} | (36792, 5271, 10540) | 0.75 |
| Solar | 137 | {96, 192, 336, 720} | (36601, 5161, 10417) | 0.33 |
| Exchange | 8 | {96, 192, 336, 720} | (5120, 665, 1422) | 0.41 |
| WCN | 90 | {96, 192, 336, 720} | (5243, 817, 15602) | 0.29 |

## C.2 METRICS

Regarding metrics, we utilize the mean square error (MSE) and mean absolute error (MAE) for long-term forecasting.

## C.3 IMPLEMENTATION DETAILS

All experiments are conducted using PyTorch 2.5.0 with CUDA 12.0, leveraging four NVIDIA A100 40GB GPUs for computation. The model is optimized using the AdamW optimizer, with the initial learning rate selected from $\{5.0 \times 10^{-5}, 1.0 \times 10^{-4}, 2.5 \times 10^{-4}, 5.0 \times 10^{-4}, 7.5 \times 10^{-4}\}$. A cosine annealing learning rate schedule is employed throughout the training process. The embedding dimension $D$ was chosen from $\{16, 32, 64, 128, 256\}$, while the patch size $p$ was fixed at 8. The batch size was determined based on dataset size, selected from $\{4, 8, 16, 32, 64, 128\}$. Training is performed for up to 50 epochs, with an early stopping mechanism that halts training if the validation performance does not improve for 10 consecutive epochs. The mean squared error (MSE) loss function is used during training. To ensure fair comparisons, the `drop_last` option is set to `False`. The code will be made available upon publication.

