# OpenReview forum: "Dual-Stream Neural Fractional Operator for Nonstationary Multivariate Time Series Forecasting"
_ICLR.cc/2026/Conference — ICLR 2026 Conference Withdrawn Submission_

### Official Review · Reviewer_XkDP · 2025-10-16

**Soundness:** 2
**Presentation:** 2
**Contribution:** 1
**Rating:** 2
**Confidence:** 4

**Summary:**

The paper proposes DualFrac, a dual-stream architecture built from Neural Fractional Operators (NFOs) that operate in the fractional Fourier domain (FrFT). Each block has a static (data-independent) and a dynamic (input-adaptive) NFO; their outputs are projected into sub-forecasts and aggregated in a residual cascade to form the final prediction. The authors provide preliminaries on FrFT/pseudo-differential operators and state approximation/convergence theorems for cascaded NFOs. Experiments on 9 benchmarks (ETT, Weather, ECL, Traffic, Exchange, Solar) and a synthetic WCN dataset claim SOTA with extensive ablations.

**Strengths:**

1. Targets non-stationarity with a principled time–frequency view; attempts to connect design with FrFT/PDO theory.

2. Includes a broad baseline table and some ablations/operator replacements; shows a lookback sweep figure (though not used to tune baselines).

3. The proposed sub-forecasting decomposition is interesting and appears promising for capturing heterogeneous temporal patterns.

**Weaknesses:**

1. There is no working anonymous code or pseudo code; the paper states only "code will be made available upon publication," which is insufficient for review.

2. Results tables (Table 3) contain suspicious entries (e.g., Exchange-720 shows several odd values like **0.1095/0.781** and **0.1092/0.769** that look like typos or fabricated result), and extreme outliers for some baselines. These inconsistencies undermine confidence in the evaluation.

3. Theorems concern approximation/cascaded convergence in FrFT/PDO spaces, but the connection to the implemented finite-depth architecture and to observed gains is not empirically validated (e.g., no diagnostics that NFO sub-forecasts align with the hypothesized non-stationary components beyond a toy example).

4. Main results fix lookback $L=96$ for all baselines, but the paper itself shows that performance varies substantially with lookback (Fig. 3), yet baselines are not tuned accordingly. A fair protocol should search input length per method and report mean±std over multiple seeds.

5. The paper’s central contribution—decomposing the prediction into sub-forecasts and aggregating them—has been extensively explored by **N-BEATS** [1], **LiNo** [2], and **Minusformer** [3]. The present manuscript seems to ***layer a fractional-domain filter (via static/dynamic NFOs) onto that established template***. However, it does not articulate what is technically new beyond this composition, nor why prior spectral/fractional operators (e.g., AFNO/FreMLP/DeepFrFT-style blocks) cannot realize the same effect. As written, ***the novelty appears incremental and primarily architectural glue.***

*[1]. N-BEATS: Neural basis expansion analysis for interpretable time series forecasting*

*[2]. LiNo: Advancing Recursive Residual Decomposition of Linear and Nonlinear Patterns for Robust Time Series Forecasting*

*[3]. Minusformer: Improving Time Series Forecasting by Progressively Learning Residuals*

**Questions:**

See in weakness.

---

### Official Review · Reviewer_uxRd · 2025-10-22

**Soundness:** 2
**Presentation:** 2
**Contribution:** 2
**Rating:** 2
**Confidence:** 5

**Summary:**

This paper proposes the Neural Fractional Operator (NFO), a learnable fractional pseudo-differential operator that performs adaptive filtering in the joint time–frequency domain, and integrates it into a dual-branch, interleaved cascade architecture to model non-stationary dynamics. Within this framework, the static branch captures global and slowly varying components, while the dynamic branch adapts to input-dependent and rapidly changing variations. Each layer cascades residuals and produces both a sub-forecast and a residual for progressive decomposition across multiple scales. The proposed model is evaluated on nine real-world benchmarks and one synthetic hyperchaotic dataset using MSE and MAE metrics, where it achieves 16 first-place and 3 second-place results across 20 comparisons, demonstrating competitive performance in forecasting complex, non-stationary time series.

**Strengths:**

1. The idea of directly modeling non-stationarity rather than relying on stationarization is insightful.

2. Extending classical fractional-operator tools with learnable symbols via hypernetworks (NFO) represents a novel and creative method for multivariate forecasting.

3. The paper includes comprehensive and systematic ablation studies.

**Weaknesses:**

1. This paper lacks either empirical or theoretical analysis (e.g., approximation guarantees, error bounds, or convergence rates) that would substantiate the claim that the Neural Fractional Operator (NFO) would enhance stationarity better than existing methods.

2. The paper does not provide sufficient formal analysis to characterize the intrinsic connection between the interleaved static/dynamic branches and the underlying NFO-based decomposition principle.

3. This method is hard to reproduce if no code provided.

4. The convergence arguments rely on components occupying (approximately) disjoint frequency bands, yet the architecture/training objective introduces no explicit band-separation constraints or regularizers. The supporting evidence remains largely assumptive, rather than being enforced or validated by the learning procedure.

5. This paper does not explain how the angular parameter is defined, shared, or constrained across layers. Because the operator becomes degenerate at certain critical angular values that the theoretical framework explicitly excludes, the authors should clarify how such degenerate settings are prevented or detected in practice and provide an analysis of the model’s numerical stability with respect to this parameter during optimization.

6. There are multiple inconsistencies that collectively raise serious doubts about the credibility of the reported baseline results and, by extension, the comparative claims:
    - WaveTS (Table 3). The detailed numbers exactly match those reported in the WaveTS paper. However, the original WaveTS results were obtained by grid-searching the lookback length from 192 to 1440, whereas this paper fixes the input length at 96. Moreover, in the paper’s own lookback sensitivity experiments, WaveTS improves (lower error) with longer lookbacks. This is not plausible under the stated setup and strongly suggests the WaveTS results were not re-run under the authors’ settings.
    - TimeMixer (Table 4) and WPMixer (Table 3). A similar issue appears: the reported results are highly consistent with the original papers, which selected the best input length among [96, 192, 336, 512], while this manuscript fixes 96. Again, obtaining essentially identical numbers in a materially different protocol is not credible.
    - SimpleTS (Table 3). The detailed results match the original paper exactly, though the original used batch size 256 on multiple datasets, whereas this paper states batch size ∈ [4, 128]. Reproducing almost identical results under such different training hyperparameters is extremely unlikely.
    - Training speed (Fig. 4a). The figure shows TimeMixer/TimeMixer+ training faster than PatchTST and even iTransformer. In practice (with the original model settings), TimeMixer variants are not that fast; these timing results are counterintuitive and insufficiently substantiated.

**Questions:**

1. In Figure 4(a), why do *TimeMixer* and *TimeMixer+* exhibit shorter training times than *PatchTST* and even *iTransformer*? These results do not follow prior evidences. Please provide sufficient details for reproducibility.

2. As far as I know, the codes of  LiNO, WaveTS and TwinsFormer are not publicly available so far. Do you reproduce them all yourself? Please also provide sufficient details for reproducibility.

3. Some plots (e.g., Figure 4) do not include the proposed model *DualFrac* but instead refer to a model named *AdaFraM*. Is this a naming error or a remnant from an earlier version of the work? Please clarify.

---

### Official Review · Reviewer_fP3Q · 2025-10-23

**Soundness:** 2
**Presentation:** 4
**Contribution:** 4
**Rating:** 4
**Confidence:** 4

**Summary:**

The paper presents DualFrac, a dual-stream, cascaded forecasting framework for non-stationary multivariate time series. Each block applies Neural Fractional Operators (NFOs) in a fractional time–frequency domain to perform both inter-series and intra-series.

**Strengths:**

1. The paper includes approximation results for cascaded fractional operators and gives credibility beyond empirical gains.
2. Strong performance gain over strong baselines (e.g., TimeMixer++) on many datasets.

**Weaknesses:**

The paper shows sub-forecasts and Wigner–Ville alignment qualitatively. It would be better if the authors add systematic interpretability metrics that components remain meaningfully non-stationary.

**Questions:**

1. How's the sensitivity of hyperparameters of the proposed method? Can you provide some experimental results that support the claim?
2. Since components remain non-stationary, how does the model adapt under **abrupt regime change** or covariate shift?

---

### Official Review · Reviewer_v2yC · 2025-10-30

**Soundness:** 3
**Presentation:** 2
**Contribution:** 2
**Rating:** 4
**Confidence:** 4

**Summary:**

This paper presents DualFrac, a novel fractional domain-based neural operator framework, aiming to solve non-stationarity—the core challenge in multivariate time series prediction. The model integrates the theoretical basis of Fractional Fourier Transform (FrFT) with the adaptability of deep learning, and designs dual-stream Neural Fractional Operators and a cascaded residual structure.

**Strengths:**

- This work introduces a fractional domain perspective, taking the Fractional Fourier Transform (FrFT) as the basis. Compared with traditional Fourier transform or wavelet decomposition (which rely on fixed or rigid bases), this approach has more theoretical advantages and enables finer modeling of non-stationary characteristics.
- This work designs a cascaded residual structure that achieves iterative refinement of input signals and component extraction, without the need to force sub-components to be stationary.
- This work achieves sota results on multiple long time series forecasting benchmark datasets, which verifies its effectiveness and robustness in handling real-world non-stationary data.

**Weaknesses:**

- The "interwoven residual update" mechanism is a key, yet seemingly heuristic, structural design. The authors must provide theoretical analysis or ablation studies to prove its mathematical superiority in non-stationary signal separation compared to simpler self-residual updates.

- Since FrFT performance is highly sensitive to the fractional order $\theta$, the paper must clarify whether $\theta$ is fixed, layer-wise learnable, or dynamically generated. A fixed $\theta$ severely limits the model's adaptivity to diverse time-frequency geometries.

- Beyond the rotational property of FrFT, the authors need to demonstrate the distinct mathematical properties and interpretability of the NFO's learned symbol $a(t, \xi)$. Is its "fractional domain filtering" provably superior to standard CNN or MLP filtering?

- The NFO is fundamentally a univariate operator. The model lacks a clear strategy for explicitly capturing and modeling coupled non-stationarity across variables in multivariate time series. Applying NFO independently on channels ignores the critical challenge of dynamic inter-dependencies.

**Questions:**

Please refer to the weaknesses part.

---

### Note · Authors · 2026-01-03

I have read and agree with the venue's withdrawal policy on behalf of myself and my co-authors.